# Entry by multiple picornaviruses is dependent on a pathway that includes TNK2, WASL, and NCK1

**Hongbing Jiang\*, Christian Leung, Stephen Tahan, David Wang\***

Department of Molecular Microbiology, Pathology and Immunology, School of Medicine, Washington University, St. Louis, United States

**Abstract** Comprehensive knowledge of the host factors required for picornavirus infection would facilitate antiviral development. Here we demonstrate roles for three human genes, *TNK2*, *WASL*, and *NCK1*, in infection by multiple picornaviruses. CRISPR deletion of *TNK2*, *WASL*, or *NCK1* reduced encephalomyocarditis virus (EMCV), coxsackievirus B3 (CVB3), poliovirus and enterovirus D68 infection, and chemical inhibitors of TNK2 and WASL decreased EMCV infection. Reduced EMCV lethality was observed in mice lacking TNK2. TNK2, WASL, and NCK1 were important in early stages of the viral lifecycle, and genetic epistasis analysis demonstrated that the three genes function in a common pathway. Mechanistically, reduced internalization of EMCV was observed in TNK2 deficient cells demonstrating that TNK2 functions in EMCV entry. Domain analysis of WASL demonstrated that its actin nucleation activity was necessary to facilitate viral infection. Together, these data support a model wherein TNK2, WASL, and NCK1 comprise a pathway important for multiple picornaviruses.

**\*For correspondence:**
hongbingjiang@wustl.edu (HJ);
davewang@wustl.edu (DW)

**Competing interests:** The authors declare that no competing interests exist.

## Introduction

Picornaviruses cause a wide range of diseases including the common cold, hepatitis, myocarditis, poliomyelitis, meningitis, and encephalitis (*Melnick, 1983*). Although vaccines exist for poliovirus and hepatitis A virus, there are currently no FDA approved antivirals against picornaviruses in the United States. As obligate intracellular pathogens, viruses are dependent on the host cellular machinery to complete their lifecycle. The early stages of the virus lifecycle, including receptor binding, entry, uncoating, and initiation of replication, are the ideal targets for preventing virus infection since the virus has not yet multiplied.

Despite extensive studies (*Yamauchi and Helenius, 2013*), there remain significant gaps in our understanding of virus entry. As the family *Picornaviridae* encompasses a wide range of viruses, it is not surprising that there is diversity in the known entry mechanisms of different species. Among the picornaviruses, poliovirus entry has been the most extensively studied. While some reports suggest that poliovirus enters the cell through clathrin-mediated endocytosis and that its genome release depends on endosome acidification (*Madshus et al., 1984a*), more recent studies report that poliovirus enters cells by a clathrin-, caveolin-, flotillin-, and microtubule-independent pathway (*Brandenburg et al., 2007*). Furthermore, poliovirus entry is sensitive to inhibitors of both tyrosine kinases and actin-polymerization, although it is not known which specific tyrosine kinase(s) is/are important for poliovirus infection (*Brandenburg et al., 2007*). Coxsackie virus B3 (CVB3) entry has also been extensively studied (*Bergelson and Coyne, 2013*). In polarized epithelial cells, CVB3 binding to the co-receptor decay-accelerating factor (DAF) and the coxsackievirus and adenovirus receptor (CAR) leads to entry by caveolin-dependent endocytosis and macropinocytosis (*Coyne and Bergelson, 2006*; *Coyne et al., 2007*). In contrast to CVB3 and poliovirus, there have been few studies of EMCV entry. Vascular cell adhesion molecule 1 (VCAM-1) and the disintegrin and

metalloproteinase domain-containing protein 9 (ADAM9) are reported to be entry factors for EMCV (*Huber, 1994*; *Bazzone et al., 2019*; *Baggen et al., 2019*). Interaction of the EMCV virion with VCAM-1 is believed to induce a conformational change that then releases the viral RNA genome; entry into the cytosol is reported to be independent of acidification (*Madshus et al., 1984b*).

Using a novel virus infection system comprised of the model organism *C. elegans* and Orsay virus, the only known natural virus of *C. elegans*, we previously identified several genes that are essential for virus infection in *C. elegans* (*Jiang et al., 2017*). The genes *sid-3*, *viro-2* and *nck-1* were found to be essential for an early, pre-replication step of the Orsay virus lifecycle. *sid-3* encodes a non-receptor tyrosine kinase orthologous to human Tyrosine Kinase Non-Receptor 2 (TNK2), *viro-2* encodes an orthologue of human Wiskott-Aldrich Syndrome protein Like protein (WASL), and *nck-1* encodes an orthologue of Non-Catalytic Region of Tyrosine Kinase (NCK1), an adaptor protein that binds to both TNK2 and WASL (*Galisteo et al., 2006*; *Donnelly et al., 2013*). Since Orsay virus is a non-enveloped, positive strand RNA virus that is evolutionarily related to the family *Picornaviridae*, we reasoned that the human orthologues of these genes may have a conserved role in infection by picornaviruses.

Human TNK2 is linked to cancers, has been reported to be activated by multiple extracellular stimuli, and is involved in many different pathways, such as clathrin/receptor mediated endocytosis, regulation of EGFR degradation, transduction of signals into the nucleus, and regulation of actin polymerization (*Galisteo et al., 2006*; *Liu et al., 2010*; *Mahajan et al., 2005*; *Yokoyama et al., 2005*). Furthermore, overexpression of TNK2 induces interferon and leads to reduced replication of a Hepatitis C virus replicon (*Fujimoto et al., 2011*). To date, there are no publications demonstrating a positive role for TNK2 in virus infection; several siRNA-based screens suggest that TNK2 is important for influenza A virus (IAV), vesicular stomatitis virus (VSV), and hepatitis C virus (HCV) infections (*Fujimoto et al., 2011*; *König et al., 2010*; *Karlas et al., 2010*; *Lupberger et al., 2011*), but no validation of these screening results have been reported. Humans encode two orthologues of *viro-2*, WASP and WASL (Wiskott-Aldrich syndrome protein /-like) (*Massaad et al., 2013*). Strikingly, biochemical assays have demonstrated that WASL is a substrate for the kinase activity of TNK2 (*Yokoyama et al., 2005*), suggesting that the two function in a pathway. NCK1 is an adaptor protein reported to co-localize with TNK2 (*Galisteo et al., 2006*; *Teo et al., 2001*) and interact with WASL. There are only limited studies linking WASL and NCK1 to virus infection. A clear role has been established for WASL in spread of vaccinia virus (*Frischknecht et al., 1999*) that involves NCK1 as well (*Donnelly et al., 2013*; *Dodding and Way, 2009*). Furthermore, sensitivity of Lassa virus infection to Wiskostatin, a small molecule inhibitor of WASL, suggests a role of WASL in virus entry (*Oppliger et al., 2016*). However, there are no studies implicating NCK1 in any aspect of picornavirus infection or in entry of any virus.

Here, we determined whether these three human orthologues of the genes found in a *C. elegans* forward genetic screen function in an evolutionarily conserved manner to facilitate virus infection in human cell culture. CRISPR-Cas9 genome editing was used to generate knockout cells for each gene, and their impact on infection by a panel of viruses from different families was then tested. Significant reductions in infection by multiple picornaviruses were observed in each of the cell lines. Consistent with these in vitro data, we observed increased survival of mice lacking murine TNK2 after EMCV challenge in vivo. We further demonstrated that TNK2, WASL, and NCK1 function in a pathway to support EMCV virus infection with WASL and NCK1 lying downstream of TNK2. Mechanistically, loss of TNK2 led to reduced EMCV virus internalization, while both TNK2 and WASL were required for proper endocytic trafficking of EMCV. These data support a model wherein TNK2, WASL, and NCK1 comprise a pathway that is important for entry by multiple picornaviruses.

## Results

### Non-receptor tyrosine kinase TNK2 is important for infection by multiple picornaviruses

To investigate the function of TNK2 in virus infection, human lung epithelial carcinoma A549 cells deficient in TNK2 were generated by CRISPR-Cas9 genome editing (*Figure 1A*). We tested a panel of viruses, including multiple viruses from the family *Picornaviridae*, which are evolutionarily related to Orsay virus (*Wolf et al., 2018*). In a single step growth analysis, deletion of *TNK2* by two

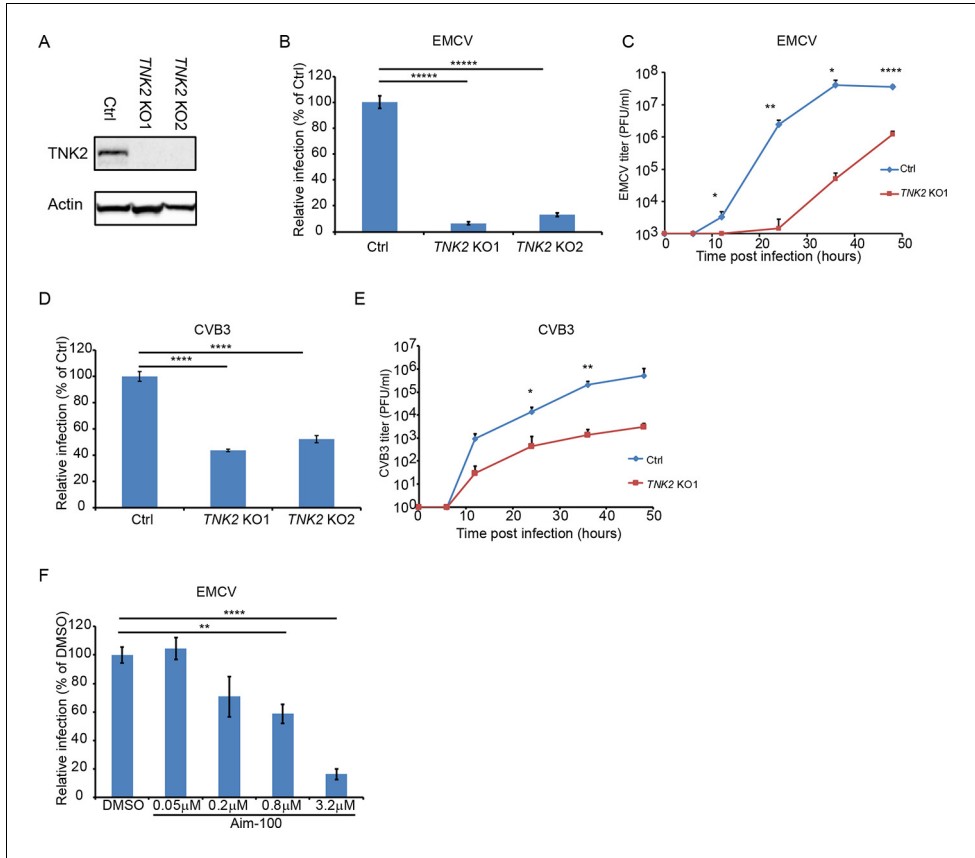

**Figure 1.** TNK2 is critical for multiple picornavirus infections. (**A**) TNK2 protein expression in *TNK2* KO1, *TNK2* KO2, and Ctrl (control) cells generated by CRISPR-Cas9 genome editing with either specific targeting or non-specific targeting sgRNA in A549 cells. Cells lysates were analyzed by Western blot. (**B**) FACS quantification of EMCV positive cells for *TNK2* KO1, *TNK2* KO2, and Ctrl cells 10 hr post infection at an MOI of 1. (**C**) Multi-step growth curve for EMCV multiplication on *TNK2* KO1 and Ctrl cells at an MOI of 0.01. Virus titers in the culture supernatant were quantified by plaque assay at 0, 6, 12, 24, 36, and 48 hr post infection. (**D**) FACS quantification of CVB3 virus positive cells for *TNK2* KO1, *TNK2* KO2, and Ctrl cells 8 hr post infection at an MOI of 1. (**E**) Multi-step growth curve for CVB3 multiplication on *TNK2* KO1 and Ctrl cells at an MOI of 0.01. Virus titers in the culture supernatant were quantified by plaque assay at 0, 6, 12, 24, 36, and 48 hr post infection. (**F**) Aim-100 inhibition of EMCV infection on naïve A549 cells. A549 cells were pre-treated with Aim-100 at indicated concentrations and infected with EMCV at an MOI of 1. Virus positive cells were quantified by FACS. (**B, D, F**) Error bars represent standard deviation of three replicates. The data shown are representatives of three independent experiments. *: $p < 0.05$, ***: $p < 0.001$, ****: $p < 0.0001$, *****: $p < 0.00001$.

The online version of this article includes the following source data and figure supplement(s) for figure 1:

**Source data 1.** Source data for *Figure 1B, D and F*: FACS quantification of virus infected cells.
**Figure supplement 1.** TNK2 is critical for multiple picornavirus infections.
**Figure supplement 2.** TNK2 and WASL are critical for multiple picornavirus infection on A549 cells.
**Figure supplement 3.** TNK2 and WASL are critical for multiple picornavirus infection on both A549 and Hap1 cells.
**Figure supplement 4.** EMCV virus infection of TNK2 rescue on knockout and control cells.

independent sgRNAs (*TNK2* KO1 and *TNK2* KO2 cells) reduced EMCV infection by 93% and 86%, respectively, compared to control cells ($p < 0.00001$, *Figure 1B* and *Figure 1—figure supplement 1A*). When infected by CVB3, 56% and 48% fewer infected *TNK2* KO1 and *TNK2* KO2 cells were observed ($p < 0.0001$, *Figure 1D* and *Figure 1—figure supplement 1B*). In a multi-step growth curve, EMCV titers in the supernatant from *TNK2* KO1 were 1660-fold lower at 24 hr post infection ($p < 0.01$, *Figure 1C*) and CVB3 titers in the supernatant from the *TNK2* KO1 were 154-fold lower at 36 hr post infection ($p < 0.01$, *Figure 1E*). In addition, reduced infectivity in *TNK2* KO1 was also

observed with both recombinant GFP-EMCV and GFP-CVB3, respectively (*Figure 1—figure supplement 2A B*). Furthermore, we examined virus replication complex formation by double stranded RNA immunostaining with the commercial J2 antibody and direct electron microscopy (EM) on EMCV-infected cells. We observed fewer cells positive for double-stranded RNA and smaller size of the virus replication complex (*Figure 1—figure supplement 2C D*). To corroborate these findings, we analyzed EMCV and CVB3 infection in commercial Hap1 cells (a haploid hematopoietic-derived lymphoma cell line) deficient in TNK2. Reductions in infected cells by 24% for EMCV and 30% for CVB3 were observed (p<0.001, *Figure 1—figure supplement 3A B*). Statistically significant reductions in poliovirus infection in both Hap1 and A549 *TNK2* knockout cells and reduction in enterovirus D68 infection in A549 cells were also observed (*Figure 1—figure supplement 3C, E F*). In contrast, no effect of *TNK2* deletion was seen for IAV, parainfluenza virus (PIV5) or adenovirus 5 (*Figure 1—figure supplement 3D, G, H I*), demonstrating the apparent specificity of TNK2 for picornavirus infection.

We next attempted to rescue the EMCV infection defect in *TNK2* knockout cells by ectopic overexpression of TNK2. However, the complexity of the *TNK2* locus (three major isoforms and 22 putative transcript variants; *Zerbino et al., 2018*) presented challenges, and the reported induction of interferon by TNK2 overexpression (*Fujimoto et al., 2011*) may have further complicated interpretation of these experiments. Lentivirus expression of each of the three major TNK2 isoforms (which yielded expression levels of TNK2 much higher than the endogenous levels (*Figure 1—figure supplement 4D*)) in control A549 cells led to statistically significant reductions in EMCV infection (*Figure 1—figure supplement 4A, B C*). In addition, fluorescently tagged TNK2 expression through either direct transfection in 293 T cells or lentivirus transduction in A549 cells led to aggregates inside the cytosol (*Figure 3—figure supplement 1* and *Figure 5—figure supplement 1A*). Nevertheless, we consistently observed a small, but statistically significant increase in EMCV virus infection in *TNK2* KO1 cells transduced with the canonical isoform one as compared to the control cells (*Figure 1—figure supplement 4A*). We also used an alternative approach to rescue virus infection in *TNK2* KO1 cells by reverting the 28 bp deletion in clone *TNK2* KO1 using CRISPR-Cas9 with oligonucleotide-mediated homologous template-directed recombination (HDR); three synonymous mutations were included in the template to unambiguously identify a successful repair event (*Figure 1—figure supplement 4H*). Screening of 500 single-cell clones yielded one clone that had repaired the *TNK2* 28 bp deletion allele. However, this process also introduced an insertion of 11 nucleotides in the adjacent intron sequence 9 bp from the nearby splice acceptor (*Figure 1—figure supplement 4H*). We were able to restore low levels of TNK2 expression in the HDR repaired clone (*Figure 1—figure supplement 4G*). In this clone, EMCV infected cell percentage and virus titer increased, by 8-fold (p<0.0001, *Figure 1—figure supplement 4E*) and 9-fold (p<0.01, *Figure 1—figure supplement 4F*), respectively as compared to *TNK2* KO1 cells.

As a completely independent means of assessing the role of TNK2, we treated A549 cells with Aim-100, a small molecule kinase inhibitor with specificity for TNK2 (*Mahajan et al., 2010*). Aim-100 pre-treatment reduced EMCV infection in a dose dependent manner without affecting cell viability (*Figure 1F*, *Figure 1—figure supplement 1C* and data not shown).

## WASL, a known substrate of TNK2, is important for picornavirus infection

The human genome encodes two WASP paralogues, WASP and WASL (also known as N-WASP). In this study, we focused on WASL because WASP is not expressed in epithelial cells such as A549. Clonal WASL knockout A549 cells were generated by CRISPR-Cas9 genome editing (*Figure 2A*). In a single step growth analysis, *WASL* KO cells showed 62% reduction in EMCV-infected cells as compared to control cells (p<0.0001, *Figure 2B* and *Figure 2—figure supplement 1A*), while in a multistep growth curve, *WASL* KO cells yielded 248-fold reduced virus titers at 24 hr post infection (p<0.01, *Figure 2C*). Ectopic expression of WASL in the *WASL* KO cells rescued EMCV infection to wild type levels (*Figure 2—figure supplement 2A B*). As with TNK2, we observed fewer cells positive for double stranded RNA by immunostaining and smaller size of the virus replication complex by EM (*Figure 1—figure supplement 2C D*). Reduced levels of CVB3, poliovirus, and enterovirus D68 infection were observed in these cells (*Figure 2D and E*, *Figure 1—figure supplement 3E, F*). In addition, independent Hap1 cells deficient in WASL had reduced EMCV, CVB3 and poliovirus infection (*Figure 1—figure supplement 3A, B C*). To independently assess the role of WASL during

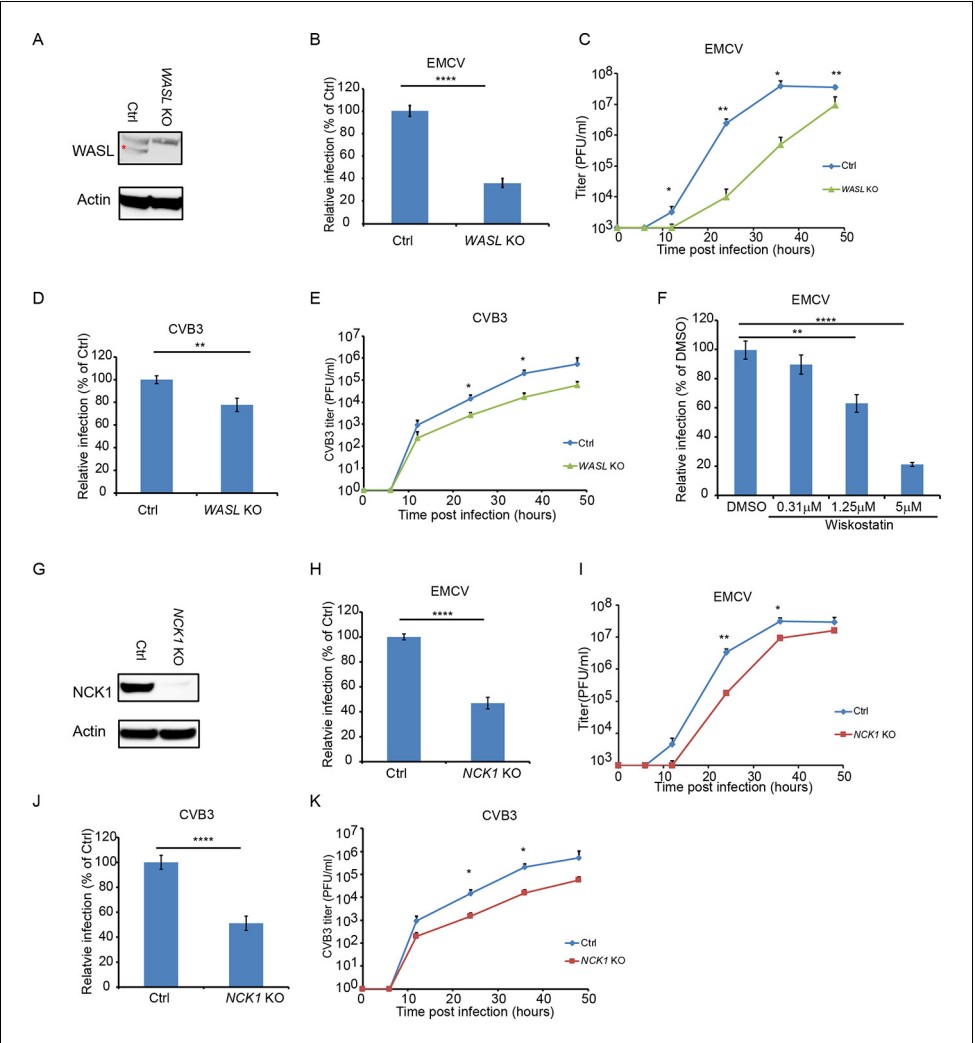

**Figure 2.** WASL and NCK1 are critical for multiple picornavirus infections. (**A**) WASL protein expression in *WASL* KO and Ctrl cells generated by CRISPR-Cas9 genome editing with either specific targeting or non-specific targeting sgRNA in A549 cells. Cells lysates were analyzed by Western blot. (**B**) FACS quantification of EMCV positive cells for *WASL* KO and Ctrl cells 10 hr post infection at an MOI of 1. (**C**) Multi-step growth curve for EMCV multiplication on *WASL* KO and Ctrl cells infected at an MOI of 0.01. (**D**) FACS quantification of CVB3 positive cells for *WASL* KO and Ctrl cells 8 hr post infection at an MOI of 1. (**E**) Multi-step growth curve for CVB3 multiplication on *WASL* KO and Ctrl cells infected at an MOI of 0.01. (**F**) Wiskostatin inhibition of EMCV infection on naïve A549 cells. A549 cells were pre-treated with Wiskotstatin at indicated concentrations and infected with EMCV at an MOI of 1. Virus positive cells were quantified by FACS. (**G**) NCK1 protein expression in *NCK1* KO and Ctrl cells generated by CRISPR-Cas9 genome editing with either specific targeting or non-specific targeting sgRNA in A549 cells. Cells lysates were analyzed by Western blot. (**H**) FACS quantification of EMCV positive cells for *NCK1* KO and Ctrl cells 10 hr post infection at an MOI of 1. (**I**) Multi-step growth curve for EMCV multiplication on *NCK1* KO and Ctrl cells infected at an MOI of 0.01. (**J**) FACS quantification of CVB3 positive cells for *NCK1* KO and Ctrl cells 8 hr post infection at an MOI of 1. (**K**) Multi-step growth curve for CVB3 multiplication on *NCK1* KO and Ctrl cells infected at an MOI of 0.01. (**A**) The red asterisk indicates WASL protein band. (**B, D, F, H, J**) Error bars represent standard deviation of three replicates. The data shown are representatives of at least two independent experiments. **: $p<0.01$, ***: $p<0.001$, ****: $p<0.0001$, *****: $p<0.00001$, NS: not significant ($p>0.05$).
The online version of this article includes the following source data and figure supplement(s) for figure 2:

**Source data 1.** Source data for *Figure 2B, D, F, H and J*: FACS quantification of virus infected cells.
**Figure supplement 1.** WASL and NCK1 are critical for multiple picornavirus infections.
**Figure supplement 2.** EMCV infection of WASL and NCK1 rescued knock out cells.

virus infection, we used a small molecule inhibitor of WASL, Wiskostatin (*Peterson et al., 2004*). Wiskostatin reduced EMCV infection in A549 cells in a dose-dependent manner with no apparent decrease of cell viability (*Figure 2F*, *Figure 2—figure supplement 1C* and data not shown). These data demonstrate that WASL is important for multiple picornavirus infection in human cell culture.

## The signaling adaptor protein NCK1 is also important for picornavirus infection

Deletion of *NCK1* in A549 cells (*Figure 2G*) reduced the number of EMCV and CVB3 infected cells by 55% and 49%, respectively (p<0.0001, *Figure 2H and J*, *Figure 2—figure supplement 1D E*) comparable to the reductions observed in *WASL* KO cells. By a multi-step growth analysis, *NCK1* KO showed 10-fold reduction of EMCV virus titer at 24 hr post infection (P<0.01 *Figure 2I*) and 14-fold reduction of CVB3 titer at 36 hr post infection (p<0.05, *Figure 2K*). Ectopic expression of NCK1 in the *NCK1* KO cells rescued EMCV infection to wild type levels (*Figure 2—figure supplement 2C D*). As with TNK2 and WASL, these data demonstrate that NCK1 is important for multiple picornavirus infection in human cell culture.

## TNK2, WASL, and NCK1 are components of a common pathway

To determine whether TNK2, WASL, and NCK1 act in a common or distinct pathway for virus infection, we performed genetic epistasis analysis by generating double and triple mutant cell lines (*Figure 3—figure supplement 2A*). EMCV infection levels and viral titer production in the double and triple knockout lines were the same as observed in the *TNK2* KO1 lines (*Figure 3A and B* and *Figure 3—figure supplement 3A*). The lack of an additive effect suggests that WASL and NCK1 are both in the same genetic pathway as TNK2. Combined with the known ability of TNK2 to phosphorylate WASL (*Yokoyama et al., 2005*), these data support a model where TNK2 acts upstream of WASL and NCK1. The magnitude of the impact of *TNK2* KO1 was greater than that of *WASL* KO or *NCK1* KO, suggesting that TNK2 might work through additional pathways in mediating virus infection besides acting through WASL and NCK1.

Because TNK2 phosphorylation of WASL increases its actin nucleation activity (*Yokoyama and Miller, 2003*), we reasoned that it might be possible to complement the TNK2 deficiency by overexpression of WASL. Overexpression of wild type WASL in the *TNK2* KO1 A549 cells led to increased EMCV virus infection (*Figure 3C*). Since constitutively active point mutants of WASL have been described (*Adamovich et al., 2009*; *Keszei et al., 2018*) (*Figure 3—figure supplement 2B C*), we next tested whether overexpression of three such constructs in the *TNK2* KO1 cells could further increase virus infection. A higher level of complementation for all three mutants was observed compared to wild type WASL (*Figure 3C*, *Figure 3—figure supplement 3B*). For NCK1, based on its reported binding to WASL and TNK2, we hypothesized that its function is to recruit WASL to TNK2, which could then activate WASL via phosphorylation (*Yokoyama et al., 2005*; *Rohatgi et al., 2001*). Thus, we also tested whether constitutively active WASL could complement *NCK1* KO. Constitutively active WASL fully rescued the *NCK1* KO virus infection phenotype (*Figure 3D* and *Figure 3—figure supplements 2D*, *3C*). Together, these data demonstrated that TNK2, WASL, and NCK1 are in a pathway to support picornavirus infection with WASL and NCK1 downstream of TNK2.

We further evaluated the interactions between TNK2, WASL and NCK1 using biochemical and biophysical assays. Previous studies had reported interaction between NCK1 and WASL through far Western and pull-down assay (*Donnelly et al., 2013*) and co-localization of TNK2 and NCK1 through protein overexpression (*Galisteo et al., 2006*). Fluorescent protein tagged forms of TNK2, WASL, and NCK1 were individually expressed in 293 T cells. WASL and NCK1 both distributed homogenously throughout the cytoplasm, while TNK2 formed puncta in the cytoplasm (*Figure 3—figure supplement 1*), which agreed with previous observations (*Teo et al., 2001*; *Shen et al., 2011*). To examine potential co-localization, we co-expressed either mCerulean tagged TNK2 or mCerulean tagged WASL with mVenus tagged NCK1 in 293 T cells, respectively. Since NCK1 is an adaptor protein, we reasoned that its interactions with binding partners would likely be relatively stable and therefore more readily detectable. Co-expression of mCerulean tagged TNK2 with mVenus tagged NCK1 re-localized NCK1 into the puncta observed by TNK2 expression alone (*Figure 3—figure supplements 1*, *4A*). Co-expression of mCerulean tagged WASL with mVenus tagged NCK1 showed homogenous distribution in the cytoplasm and co-localization of these two proteins (*Figure 3—*

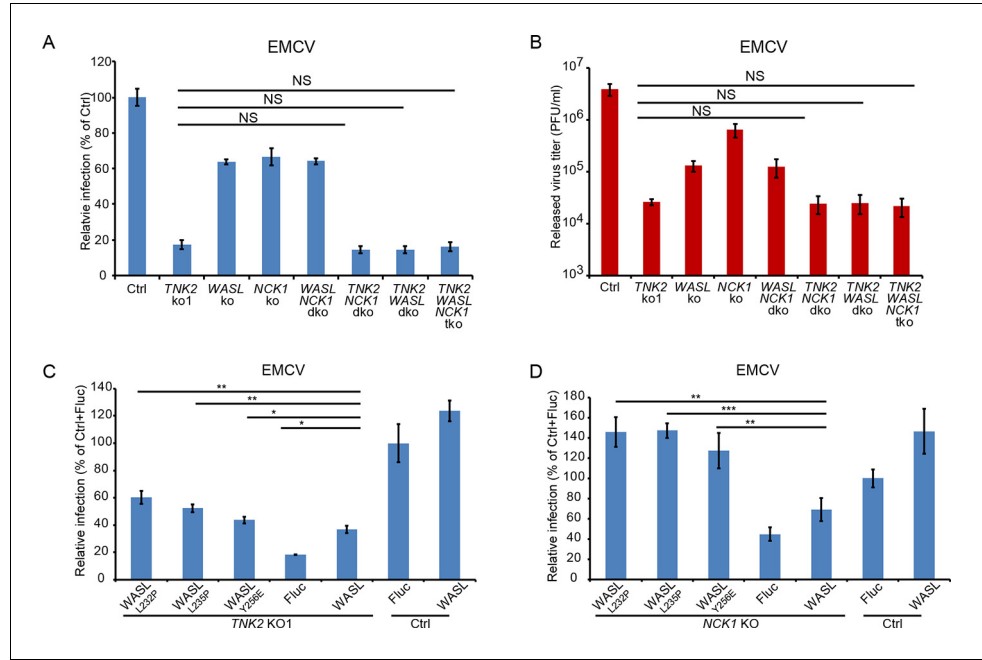

**Figure 3.** TNK2, WASL, and NCK1 are in a pathway supporting virus infection. (**A**) FACS quantification of EMCV positive cells for *TNK2*, *WASL*, *NCK1* single, double, triple gene knockout and Ctrl cells 10 hr post infection at an MOI of 1. (**B**) Virus titer for EMCV multiplication on *TNK2*, *WASL*, and *NCK1* single, double, triple gene knockout and Ctrl cells at 24 hr post infection at an MOI of 0.01. (**C**) FACS quantification of EMCV positive cells for *TNK2* KO1 cells that were transduced with constitutively active WASL constructs 10 hr post infection at an MOI of 1. (**D**) FACS quantification of EMCV positive cells for *NCK1* KO cells that were transduced with constitutively active WASL constructs 10 hr post infection at an MOI of 1. (**A, B**) dko: double knockout, tko: triple knockout. (**A–D**) Error bars for virus infection represent standard deviation of three replicates. The data shown are representatives of two independent experiments.

The online version of this article includes the following source data and figure supplement(s) for figure 3:

**Source data 1.** Source data for *Figure 3A, C and D*: FACS quantification of virus infected cells.
**Figure supplement 1.** Co-localization of fluorescently tagged NCK1, TNK2, and WASL expressed in 293 T cells by confocal imaging.
**Figure supplement 2.** Gene expression in knock out cells and constitutively active WASL expression.
**Figure supplement 3.** TNK2, WASL, and NCK1 are in a pathway supporting virus infection.
**Figure supplement 4.** TNK2 and WASL directly interact with NCK1.

---

*figure supplements 1*, *4A*). Further analysis by quantitative FRET demonstrated that WASL and NCK1 had higher FRET efficiency in cells, while TNK2 and NCK1 also had significant, but lower, FRET efficiency (*Figure 3—figure supplement 4A B*). Consistent with the FRET data, in co-transfected 293 T cells, immunoprecipitation of FLAG tagged NCK1 pulled down HA tagged WASL more efficiently than it did Myc tagged TNK2 (*Figure 3—figure supplement 4C*). Altogether, these data demonstrate that TNK2 and WASL bind directly to NCK1.

## TNK2, WASL, and NCK1 affect a pre-replication step of the EMCV lifecycle

To begin dissecting the stage of the EMCV virus lifecycle impacted by TNK2, WASL, and NCK1, we transfected EMCV genomic RNA into the cells to bypass the early stages of the viral lifecycle. The *TNK2* KO1, *WASL* KO, and *NCK1* KO cells produced the same titers of EMCV virus as the control cells at 10 hr post transfection (*Figure 4A*) demonstrating that TNK2, WASL, and NCK1 are dispensable for EMCV replication and post-replication stages of the EMCV lifecycle. Thus, we concluded that all three genes act on an early stage of the viral lifecycle. As a second line of evidence, we determined the impact of adding the TNK2 kinase-specific inhibitor Aim-100 before or at varying times after EMCV infection. While Aim-100 pre-treatment reduced EMCV infection, administration

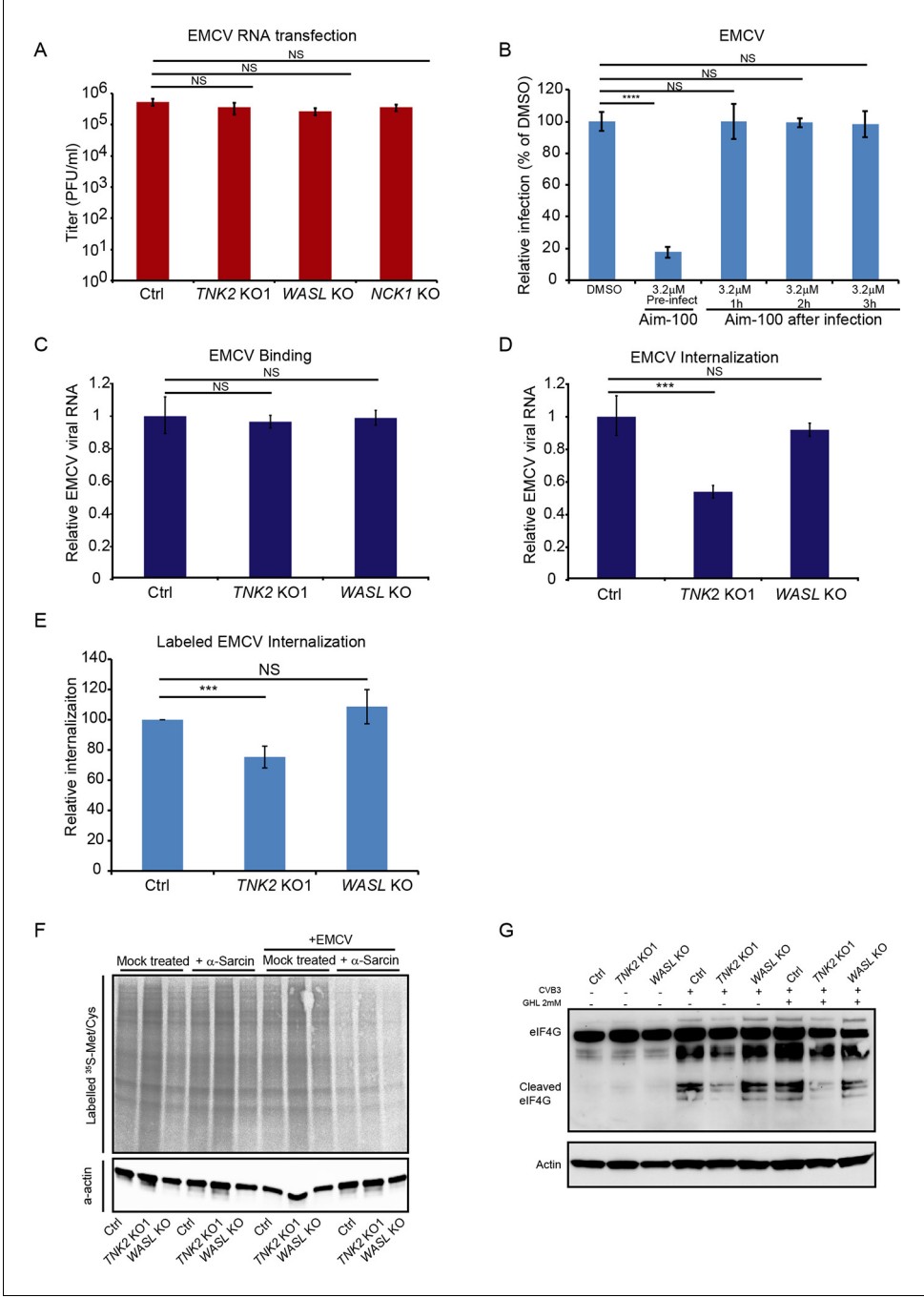

**Figure 4.** TNK2, WASL, and NCK1 function at an early stage of virus lifecycle. (**A**) EMCV released from viral RNA transfected Ctrl, *TNK2* KO1, *WASL* KO, and *NCK1* KO cells 10 hr post transfection was quantified by plaque assay. (**B**) Time-dependent addition of Aim-100 on EMCV infection on naïve A549 cells. A549 cells were treated with 3.2 µM Aim-100 at different time points before and after EMCV infection at an MOI of 1. EMCV positive cells were then quantified by FACS. (**C**) Quantification of EMCV virus binding on *TNK2* KO1 and *WASL* KO cells by qRT-PCR expressed as relative change to Ctrl cell binding. (**D**) Quantification of EMCV virus internalization in *TNK2* KO1 and *WASL* KO cells by qRT-PCR expressed as relative change to Ctrl cell internalization. (**E**) FACS quantification of labeled EMCV internalization in *TNK2* KO1, *WASL* KO and Ctrl cells. (**F**) a-sarcin pore forming assay performed on *TNK2* KO1, *WASL* KO and Ctrl cells. Translation was measured by phosphorimaging of $^{35}$S-methionine/cysteine incorporation. (**G**) eIF4G cleavage by CVB3 infection for 2 hr with or without 2 mM guanidine hydrochloride. (**A–E**) Error bars represent standard deviation of three replicates. The data shown are representative of two independent experiments. ***: $p<0.001$, ****: $p<0.0001$, NS: not significant ($p>0.05$).

*Figure 4 continued on next page*

*Figure 4 continued*

The online version of this article includes the following source data and figure supplement(s) for figure 4:

**Source data 1.** Source data for *Figure 4B*: FACS quantification of virus infected cells.
**Figure supplement 1.** TNK2, WASL, and NCK1 function at an early stage of virus lifecycle.
**Figure supplement 2.** EMCV internalization in Ctrl, *TNK2* KO1 and *WASL* KO cells.
**Figure supplement 3.** Transferrin, and dextran uptake in Ctrl, *TNK2* KO1, and *WASL* KO cells.

after virus inoculation had no effect (*Figure 4B* and *Figure 4—figure supplement 1*), consistent with a role for TNK2 at an early stage of virus infection.

## TNK2, but not WASL functions in virus internalization

We examined the impact of *TNK2* and *WASL* deletion on virus binding, virus internalization, pore formation, and viral RNA genome translocation. Following incubation at 4°C for one hour, no difference in virus binding was observed among control, *TNK2* KO1, and *WASL* KO cells (*Figure 4C*), as assessed by qRT-PCR for EMCV genomic RNA. In contrast, after raising the temperature to 37°C for 30 min to allow for internalization of the bound virus particles and then trypsinization to remove uninternalized surface virus particles as described before (*Berry and Tse, 2017*; *Hackett et al., 2015*), there was a 46% reduction (p<0.001) of intracellular EMCV viral RNA in the *TNK2* KO1 cells (*Figure 4D* and *Figure 4—figure supplement 2C–E*). However, no difference in RNA levels was observed in *WASL* KO cells (*Figure 4D*). As a complementary approach, we evaluated internalization using fluorescently labeled EMCV virions (*Figure 4—figure supplement 2A B*). 25% fewer EMCV positive *TNK2* KO1 cells were observed compared to control cells (*Figure 4E* and *Figure 4—figure supplement 2B*), while no statistical difference was observed in the *WASL* KO cells (*Figure 4E* and *Figure 4—figure supplement 2B*).

We examined the ability of fluorescently labeled transferrin, a known cargo for clathrin-mediated endocytosis, to be endocytosed in both *TNK2* KO1 and *WASL* KO cells. Transferrin was internalized in *TNK2* KO1 and *WASL* KO cells to the same level as in the control cells (*Figure 4—figure supplement 3A B*), consistent with a previous report that knock down of TNK2 has no effect on transferrin endocytosis (*Grøvdal et al., 2008*). To further check other general virus entry pathways, we also examined whether macropinocytosis was affected in these cells. FITC conjugated Dextran (7 KDa) is reported to be uptaken by macropinocytosis (*Oppliger et al., 2016*). No defect in uptake of Dextran was observed in *TNK2* KO1 cells (*Figure 4—figure supplement 3C D*). In contrast, the *WASL* KO cells had a 75% reduction in Dextran uptake (*Figure 4—figure supplement 3C D*), which is consistent with the known dependency of macropinocytosis on actin polymerization (*Innocenti et al., 2005*). The Dextran macropinocytosis defect in the *WASL* KO appears to be unrelated to EMCV infection though since no reduction in either binding or internalization of EMCV was observed.

A step that is closely linked to internalization for picornaviruses is virus-induced pore formation, which is required for genome release into the cytoplasm. To assess pore formation, cells were treated with the membrane impermeable translation inhibitor, α-sarcin. Virus-induced pore formation will lead to translation inhibition and reduction of $^{35}$S methionine and $^{35}$S cysteine incorporation whereas in the absence of pore formation, translation will proceed at wild type levels (*Fernández-Puentes and Carrasco, 1980*). *TNK2* KO1 and *WASL* KO cells showed no defect in virus pore formation during EMCV infection (*Figure 4F*). Although a clear defect in EMCV internalization exists in the *TNK2* KO1 cells (*Figure 4D and E*), no defect in pore formation was observed. Since the internalization defect in *TNK2* KO1 cells is not absolute, pore formation may still occur with sufficient frequency to inhibit translation. Alternatively, pore formation may not necessarily require prior internalization; for example binding of parainfluenza virus alone is sufficient to initiate pore formation at the cell membrane (*Porotto et al., 2012*). For WASL, no defect in binding, internalization or pore formation was observed in the knockout cells, suggesting that the defect is downstream of these steps. Similarly, a recent report suggests that the human gene PLA2G16 acts at a step of the picornavirus lifecycle that is subsequent to pore formation (*Staring et al., 2017*).

To determine whether TNK2 or WASL plays any role in downstream viral RNA genome translocation after pore formation, we examined virus-induced host protein cleavage after its entry. For some picornaviruses, immediately after translocation of its genome, translation of the incoming positive

strand RNA leads to proteolytic cleavage of the host protein eIF4G to shut down host translation (*Glaser and Skern, 2000*). Since this activity has not been described for EMCV, we used CVB3 which is known to have this activity (*Carthy et al., 1998*). eIF4G cleavage in presence or absence of a virus genome replication inhibitor guanidine hydrochloride showed no difference in the *WASL* KO cells as compared to the control cells, demonstrating that WASL does not impact this step of the viral life-cycle. In contrast, we observed a decrease of eIF4G cleavage in *TNK2* KO1 cells, suggesting that TNK2 might play a role in this process. Alternatively, the reduced eIF4G cleavage level might simply reflect the reduced level of virus internalization in *TNK2* KO1 cells (*Figure 4G*).

## TNK2 was found in proximity to labeled EMCV particles during infection

TNK2 is present in endocytic vesicles and colocalizes with the early endosome marker EEA1 (*Teo et al., 2001*; *Shen et al., 2011*; *Grøvdal et al., 2008*; *Jones et al., 2014*). To visualize TNK2 subcellular localization, we expressed N-terminally GFP tagged TNK2 in the *TNK2* KO1 cells. Two different patterns were observed in GFP positive cells: high TNK2 expression formed GFP aggregates inside the cytosol of cells in about 80% of the cell population, which has been described previously (*Teo et al., 2001*; *Prieto-Echagüe et al., 2010*), and low TNK2 expression inside the cytosol of cells in about 20% of the cell population (*Figure 5—figure supplement 1A, B and C*). Consistent with the native TNK2 rescue, N-terminally GFP tagged TNK2 expression in the *TNK2* KO1 cells statistically increased virus infection from 17.2% to 22.0% of GFP-transduced control cell infection (p<0.01, *Figure 5—figure supplement 1E*). The modest increase was comparable to that observed with ectopic expression of wild type TNK2. Next, we checked TNK2 localization in cells infected with fluorescently labeled EMCV. In *TNK2* KO1 cells with high TNK2 expression that formed aggregates, no GFP-TNK2 in proximity to EMCV was observed (*Figure 5—figure supplement 1A*). However, in cells with low TNK2 expression, labeled EMCV particles were observed in proximity to GFP-TNK2 (*Figure 5A* and *Figure 5—figure supplement 1B*), presumably in early endosomes. Further quantification demonstrated that about 40% of the EMCV particles were in proximity to GFP-TNK2 in those cells (*Figure 5—figure supplement 1D*).

WASL is recruited to vaccinia virus vesicles to promote its intracellular trafficking (*Dodding and Way, 2009*). To examine whether EMCV infection recruits WASL as observed for poxvirus, we expressed N-terminal GFP tagged WASL in *WASL* KO cells. The GFP tagged WASL was functional as it rescued EMCV infection (*Figure 5—figure supplement 1F*). Following infection with fluorescently labeled EMCV, no clear recruitment of GFP-WASL to fluorescently labeled EMCV was observed (*Figure 5A*).

## EMCV infection is CDC42 dependent but clathrin independent

Both TNK2 and WASL are activated by the upstream CDC42 small GTPase (*Galisteo et al., 2006*; *Grøvdal et al., 2008*; *Howlin et al., 2008*; *Mahajan et al., 2007*; *Rohatgi et al., 2000*), which can be inhibited by pirl1 (*Oppliger et al., 2016*). Pirl1 treatment inhibited EMCV replication at a 10 µM concentration without cell toxicity (*Figure 5B*, *Figure 5—figure supplement 2* and data not shown), suggesting that CDC42 mediates EMCV infection.

Since a role for TNK2 in clathrin-mediated endocytosis (CME) has been reported (*Teo et al., 2001*), we next examined whether the classic CME inhibitors such as dynasore (a canonical inhibitor of dynamin) or pitstop-2 (an inhibitor of clathrin) affect EMCV virus infection. Dynasore and pitstop-2 had no effect on EMCV infection while they inhibited VSV, a virus known to rely predominantly on CME for its entry, in a dose dependent fashion (*Figure 5—figure supplement 3A B*). Together, these data indicate that EMCV infection in A549 cells is dependent on CDC42, which activates TNK2 and WASL, but does not require CME for its entry.

## EMCV virions accumulate in early endosomes in the absence of TNK2 or WASP

We next examined colocalization of EMCV with the early endosome marker EEA1. A time course analysis demonstrated that in control cells, the peak colocalization of EMCV with EEA1 occurred at 20 min post internalization (*Figure 5—figure supplement 3C D*), which was followed by a decrease at 30 min post internalization. This result is consistent with a previous report that poliovirus entry

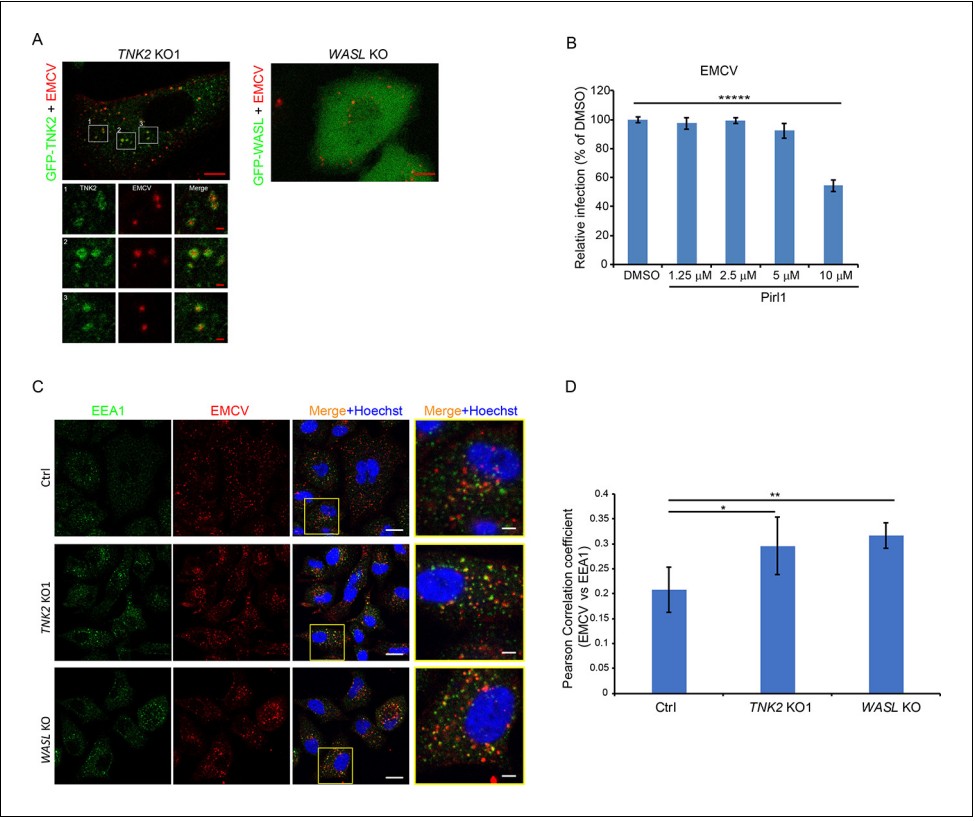

**Figure 5.** TNK2 mediates virus infection through endosomal trafficking pathways. (**A**) Confocal imaging of GFP-tagged TNK2 localization with fluorescently labeled EMCV virus in *TNK2* KO1 cells and GFP-tagged WASL localization with fluorescently labeled EMCV virus in *WASL* KO cells. Scale bars represent 10 μm. Individual channels of different insets were shown. Scale bars represent 2 μm. (**B**) FACS quantification of EMCV infection on pirl1 treated A549 cells at 10 hr post infection at an MOI of 1. Error bars represent standard deviation of three replicates. (**C**) EEA1 staining of fluorescently labeled EMCV infected Ctrl, *TNK2* KO1, and *WASL* KO cells. Scale bars represent 20 μm. Insets represent magnification of the boxed region. Scale bars represent 5 μm. (**D**) Quantification of Pearson correlation coefficient of EMCV and EEA1 colocalization in Ctrl, *TNK2* KO1, and *WASL* KO cells infected with fluorescently labeled EMCV. 33 cells for Ctrl, 37 cells for *TNK2* KO1 and 19 cells for *WASL* KO were quantified. Error bars represent standard deviation of images quantified. (**B, D**) The data shown are representative of two independent experiments. *: $p < 0.05$, **: $p < 0.01$.

The online version of this article includes the following source data and figure supplement(s) for figure 5:

**Source data 1.** Source data for *Figure 5B*: FACS quantification of virus infected cells.
**Figure supplement 1.** Localization of GFP-TNK2 with fluorescently labeled EMCV.
**Figure supplement 2.** TNK2 mediates virus infection through endosomal trafficking pathways.
**Figure supplement 3.** The endocytosis pathway inhibition on EMCV infection and localization of EEA1 with fluorescently labeled EMCV.

peaks at 20 min post infection (*Brandenburg et al., 2007*). In contrast, in both the *TNK2* KO1 and *WASL* KO cells, higher levels of EMCV particles were retained in EEA1 containing vesicles at 30 min post internalization (*Figure 5C, D*, and *Figure 5—figure supplement 3D*). Thus, both *TNK2* KO1 and *WASL* KO cells are characterized by increased accumulation of EMCV particles in early endosomes.

## EMCV infection is dependent on the activation of WASL and its actin modulating function

The primary known effector function of WASL is nucleation of actin polymerization, which is mediated by binding of its C-terminal domain to the Arp2/3 complex and actin monomers (*Galletta et al., 2008*). Different domains of WASL are critical for interaction with other proteins that

can modulate WASL stability, activation, or function. Because ectopic expression of wild type WASL fully rescued EMCV infection in *WASL* KO cells, we tested the ability of a series of WASL domain deletion mutant constructs to rescue infection in *WASL* KO cells (*Figure 6A*). The basic region that binds with PIP2, the proline rich region that binds with SH3 domain containing proteins, the GTPase binding domain that interacts with CDC-42 kinase, and the acidic domain that is involved in actin binding and polymerization were all required for EMCV infection (*Figure 6B and C* and *Figure 6— figure supplement 1A*). Interestingly, transduction of the N-terminal WH1 domain truncated mutant increased EMCV infection by 3-fold compared to wild type WASL transduction (*Figure 6B* and *Figure 6—figure supplement 1A*). The WH1 domain is reported to bind with WASP interacting proteins (WIPs) to maintain WASL in an inactivated state (*Stradal et al., 2004*); therefore, the WH1 domain truncation mutant should have increased actin polymerization activity. To further assess the WASL pathway's role in EMCV infection, we treated cells with the Arp2/3 complex inhibitor CK-869. CK-869 treatment inhibited EMCV infection in a dose-dependent manner without cell toxicity (*Figure 6D*, *Figure 6—figure supplement 1B* and data not shown). Altogether, these data demonstrated that WASL and its downstream actin pathway play important roles in EMCV infection.

## TNK2 is required for EMCV infection in an in vivo mouse model

To investigate TNK2's role in EMCV infection in vivo, we generated a *Tnk2* knockout mouse that carried a 13 Kb deletion of the murine *Tnk2* genomic locus, which eliminated all annotated TNK2 isoforms and splice variants (*Figure 7A*). In primary mouse lung fibroblast cells, complete ablation of TNK2 protein expression was observed by Western blot (*Figure 7B*). The number of cells infected

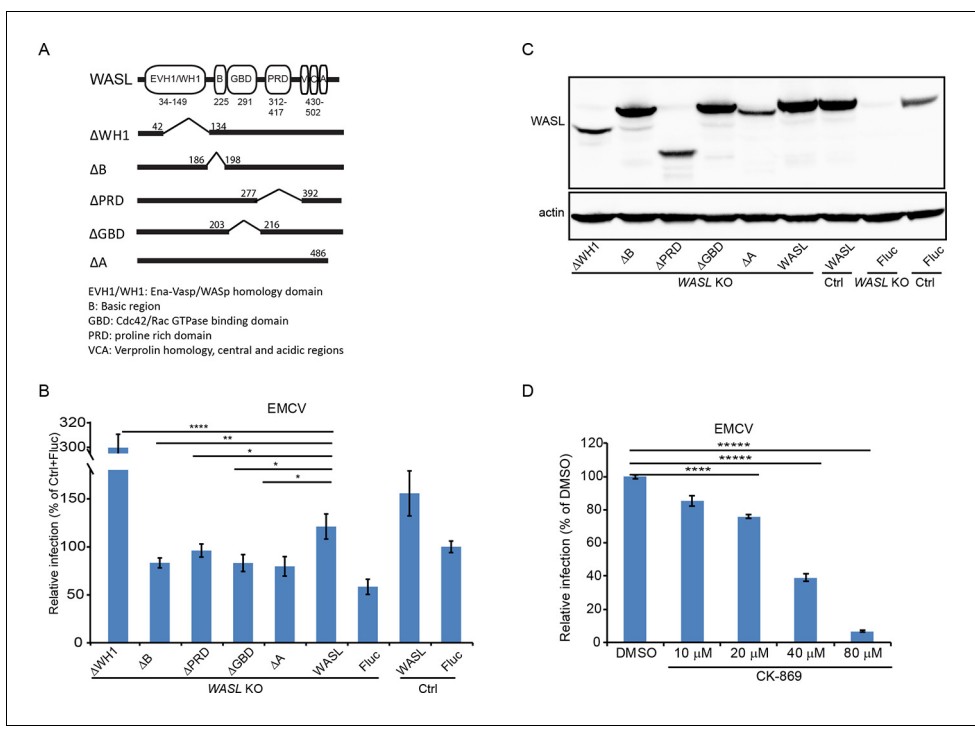

**Figure 6.** WASL activation and its actin modulation are critical for EMCV virus infection. (**A**) Schematic representation of different WASL domain truncations. Each truncation is indicated by amino acid position on the constructs. (**B**) FACS quantification of EMCV infection in *WASL* KO cells transduced with different WASL domain truncations. (**C**) Western blot detection of WASL domain truncation expression constructs in lentivirus transduced *WASL* KO cells. (**D**) CK-869 inhibition of EMCV infection on naïve A549 cells at 10 hr post infection at an MOI of 1. (**B, D**) Error bars represent standard deviation of three replicates. The data shown are representative of two independent experiments. *: p<0.05, **: p<0.01, ***: p<0.001, ****: p<0.0001.
The online version of this article includes the following source data and figure supplement(s) for figure 6:

**Source data 1.** Source data for *Figure 6B and D*: FACS quantification of virus infected cells.
**Figure supplement 1.** WASL activation and its actin modulation are critical for EMCV virus infection.

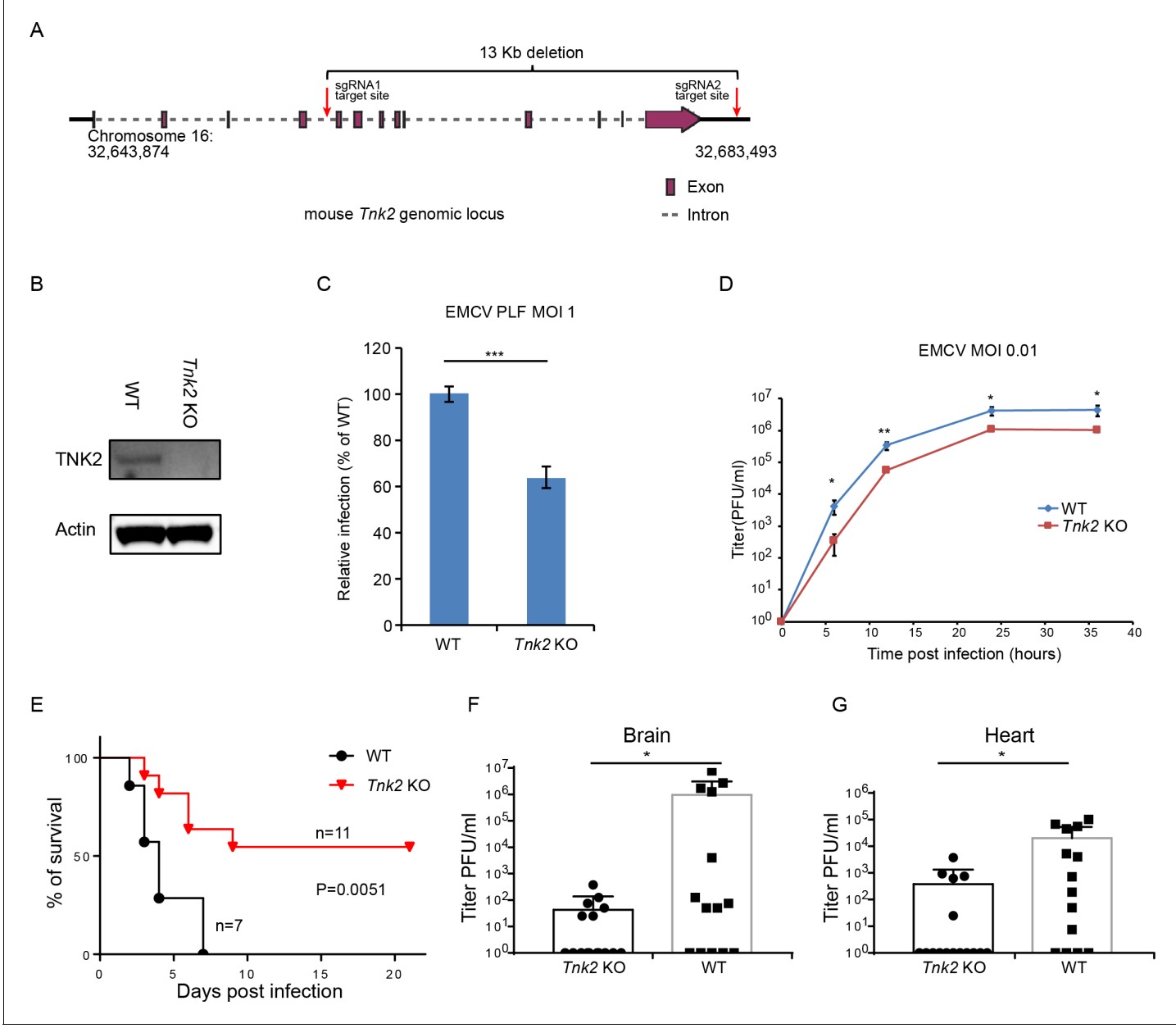

**Figure 7.** TNK2 is required for EMCV infection in vivo. (**A**) Schematic representation of *Tnk2* knockout design by CRIPSR-Cas9 genome editing in mouse. Exon, intron and genomic position are indicated. (**B**) TNK2 expression in mouse primary lung fibroblast cells derived from *Tnk2* knockout and wild type animals. Cell lysates were analyzed by Western blot. (**C**) FACS quantification of EMCV infection in mouse primary lung fibroblast cells derived from *Tnk2* knockout and wild type animals 6 hr post infection at an MOI of 1. Error bars represent standard deviation of three replicates. The data shown are representative of two independent experiments. ***: p<0.001. (**D**) Multi-step growth of EMCV in mouse primary lung fibroblast cells from *Tnk2* knock out and wild type animals. *: p<0.05, **: p<0.01. (**E**) Survival curve of EMCV infection via oral gavage in *Tnk2* knockout and wild type mice. p=0.0051 by log-rank test. (**F, G**) EMCV titer in infected mouse brain and heart. *: p<0.05 by Mann-Whitney test, n = 16 for *Tnk2* KO and n = 14 for WT. The online version of this article includes the following source data and figure supplement(s) for figure 7:

**Source data 1.** Source data for *Figure 7C, E, F, G*.
**Figure supplement 1.** TNK2 is required for EMCV infection in vivo.

by EMCV was reduced by 40% in the knockout cells (p<0.001, *Figure 7C* and *Figure 7—figure supplement 1*). A multi-step growth analysis showed that wild type primary lung fibroblast had more than 10-fold higher virus production than *Tnk2* knock out cells at 6 hr post infection (p<0.01, *Figure 7D*). In vivo, EMCV challenge with $10^7$ PFU by oral gavage resulted in greater survival of *Tnk2* knockout animals compared to wild type animals (p=0.0051, *Figure 7E*). Virus titration in infected mouse tissues showed that *Tnk2* knock out mice had four logs lower virus titer in average in brain and two logs lower virus titer in average in heart (p<0.05, *Figure 7F and G*). Together, these data demonstrated an important role for TNK2 in EMCV infection in an in vivo mouse infection model.

## Discussion

We previously determined that the *C. elegans* genes *sid-3, viro-2,* and *nck-1* are essential for Orsay virus infection in *C. elegans* (*Jiang et al., 2017*). In this study, we further asked whether their respective human orthologues *TNK2, WASL,* and *NCK1* play any roles in mammalian virus infection. After screening a panel of mammalian viruses, we found that multiple picornaviruses, including EMCV, CVB3, enterovirus D68, and poliovirus, rely on TNK2, WASL, and NCK1 for infection in different cell lines. For TNK2, this represents the first evidence that it has any pro-viral function. Interestingly, poliovirus entry was previously shown to be dependent on an unknown kinase based on the use of a broad-spectrum tyrosine kinase inhibitor (*Brandenburg et al., 2007*). One possibility is that TNK2 might be the kinase targeted in that study.

Our data represent the first demonstration that WASL and NCK1 are important for infection by picornaviruses such as EMCV, CVB3, enterovirus D68, and poliovirus. RNA transfection of EMCV into *TNK2, WASL,* and *NCK1* knockouts each yielded the same level of virus as the control demonstrating their role in an early stage of the EMCV lifecycle. For WASL, this is in contrast to its well-defined role in facilitating spread and transmission of vaccinia virus (*Frischknecht et al., 1999*) suggesting that WASL is important for different lifecycle stages for different viruses. In addition to the known interactions between NCK1 and WASL, TNK2 has been reported to phosphorylate WASL (*Yokoyama et al., 2005*). Consistent with these data, our genetic epistasis analysis and ectopic trans-complementation data (*Figure 3A–D*) demonstrated that these three genes function in a pathway in the context of picornavirus infection. Furthermore, biochemical and biophysical data demonstrated direct interaction of NCK1 with both TNK2 and WASL. Previous studies have established a clear role for CDC42 in regulation of TNK2 and WASL (*Rohatgi et al., 2000*; *Prieto-Echagüe and Miller, 2011*), and inhibition of CDC42 resulted in reduced EMCV infection.

We explored the mechanism by which TNK2 and WASL act regarding picornavirus infection. *TNK2* KO1 cells were partially defective in EMCV internalization. In contrast, internalization was not affected in *WASL* KO cells, suggesting WASL functions in one or more steps downstream of virus internalization. In the absence of either TNK2 or WASL, EMCV particles continued to accumulate in early endosomes at 30 min post infection in contrast to control cells, demonstrating a role for both genes in endosomal trafficking. Notably, *sid-3*, the *C. elegans* orthologue of TNK2, is important for endosomal trafficking of RNA molecules (*Jose et al., 2012*). Thus, one possibility is that the delay in release from the early endosomes contributes to the reduction in EMCV infection in both *TNK2* and *WASL* KO cells. However, it is uncertain whether this observed accumulation in early endosomes is directly related to the reduced levels of infection in these cells. In detailed studies of poliovirus, most of the RNA genome is released from the virus particles by 20 min post internalization (*Brandenburg et al., 2007*). Thus, additional studies are necessary to determine whether the accumulated particles in the early endosomes still contain genomic RNA.

These data led us to the following model (*Figure 8*). TNK2 is in a pathway upstream of WASL and NCK1. Because *TNK2* deletion has a greater magnitude of impact than *WASL* or *NCK1* deletion, TNK2 must act on one or more additional step of the viral lifecycle than WASL and NCK1. Consistent with this model, we demonstrated that TNK2 (but not WASL) is needed for virus internalization. In addition, we propose that TNK2 activates WASL (presumably, but not necessarily, by phosphorylation and in a fashion mediated by NCK1 binding), which then affects a subsequent early step of the EMCV lifecycle. Our data suggest that this step is linked to proper endocytic trafficking of the incoming viral particles, which accumulated in early endosomes in both TNK2 and WASL deficient cells to a greater extent than in control cells. The requirement for the WASL acidic domain, which is

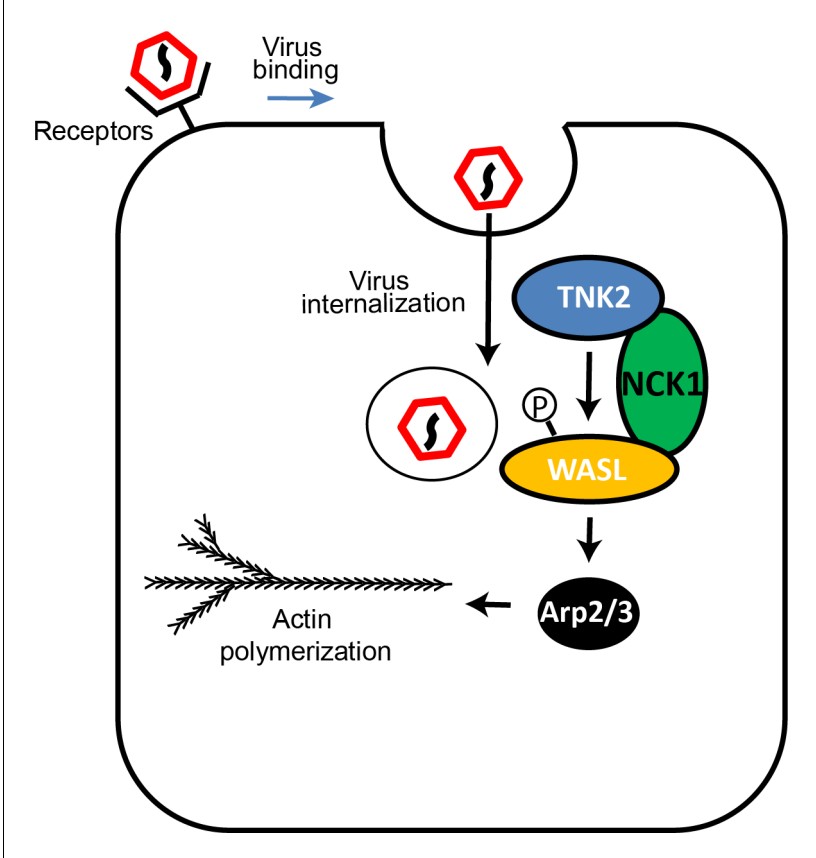

**Figure 8.** Model of TNK2, WASL, and NCK1 function in picornavirus infection.

responsible for actin binding and nucleation, along with the sensitivity to the Arp2/3 inhibitor CK-869, demonstrates that this process is actin dependent. One possibility is that regulated actin polymerization may be necessary to deliver endocytic vesicles to specific subcellular locations necessary for productive infection.

Actin is critical for many cellular processes and due to its essentiality, is not a druggable target. In this study, we identified a TNK2 and actin-dependent pathway necessary to promote virus infection but does not appear to alter normal endocytosis. As cells lacking TNK2 and NCK1 are clearly viable, as are mice lacking TNK2 and NCK1 (*Bladt et al., 2003*). These genes are potential antiviral targets that could be disrupted by small molecules without compromising overall actin biology and survival of the host.

It is becoming increasingly clear that there are additional, poorly defined stages of the picornavirus lifecycle. A recent genetic screen identified PLA2G16 as an essential gene required for an early stage of multiple picornaviruses at a step post binding, internalization, and pore formation (*Staring et al., 2017*). Our study similarly suggests that WASL also acts after these stages. Further study is needed to determine whether WASL acts at the same stage as PLA2G16 or at a distinct step.

All four picornaviruses we tested were dependent on TNK2 and WASL. In contrast, viruses in other families such as influenza A virus, parainfluenza 5 and adenovirus A5 were not affected by absence of these genes, suggesting that picornaviruses may specifically depend on these host factors. PIV5 is known to enter cells by direct fusion at the cellular membrane and is not believed to utilize endocytic vesicles (*Porotto et al., 2012*). The observed lack of dependence of PIV5 on TNK2 or WASL is consistent with a potential role of TNK2 and WASL in modulating endocytic vesicle trafficking. In contrast, picornaviruses such as poliovirus, CVB3, and EMCV are generally thought to utilize clathrin independent endocytic vesicles for entry (*Bergelson and Coyne, 2013*), while influenza A virus and adenovirus mostly rely on clathrin-mediated endocytosis for their entry (*Yamauchi and*

*Helenius, 2013*). The magnitude of the phenotypes did vary significantly among the tested picorna-viruses, which is not surprising given the significant degree of divergence between these virus genomes and differences in their entry requirements. Nonetheless, all four tested picornaviruses displayed different degree of dependence on this pathway. It will be important to test a wider range of picornaviruses to determine exactly which taxa within the *Picornaviridae* are most reliant on these genes. It will also be of great interest to test additional viruses from other families. The data suggesting a role of WASL in Lassa virus infection (*Oppliger et al., 2016*) supports the notion that at least some viruses in other virus families may also depend on one or more of these genes for infection.

In recent years, multiple studies of host factors critical for picornavirus infection have utilized haploid gene-trap screening or CRISPR screening in mammalian cell lines (*Bazzone et al., 2019*; *Staring et al., 2017*; *Kim et al., 2017*). These studies have identified multiple host genes required broadly for picornaviruses (*Staring et al., 2017*), as well as examples of genes required for specific picornaviruses. For instance, ADAM9 was discovered as a potential receptor for EMCV by CRISPR and gene trap screening (*Bazzone et al., 2019*; *Baggen et al., 2019*). Notably, the genes that we identified by starting with the *C. elegans* Orsay virus genetic screen, *TNK2*, *WASL*, and *NCK1*, were not identified as hits in any of the direct mammalian screens undertaken by others (*Bazzone et al., 2019*; *Staring et al., 2017*; *Kim et al., 2017*). Thus, these approaches provide complementary experimental strategies that, ideally in the long run, will lead to comprehensive understanding of host factors necessary for virus infection.

Our study of *TNK2* was driven by our identification in *C. elegans* of *sid-3* as a host factor required for Orsay virus infection (*Jiang et al., 2017*). *sid-3* was originally identified in a genetic screen for mutants defective in systemic spread of RNAi (*Jose et al., 2012*). That screening strategy also identified another *C. elegans* gene, *sid-1*, that is reported to transport dsRNA (*Winston, 2002*). A recent study demonstrated that the mammalian orthologue of *sid-1*, SIDT2, has an evolutionarily conserved function in transporting dsRNA that serves as a means of activating innate antiviral immunity in mice (*Nguyen et al., 2017*). Those studies provide an example of the value of using the *C. elegans* model system to guide insights into conserved functions with mammals. Here we demonstrated that TNK2, the human orthologue of *sid-3*, has an evolutionarily conserved function to facilitate virus infection in mammals. In *C. elegans*, Orsay virus infection is presumed to occur by fecal-oral transmission, as evidenced by its exclusive intestinal cell tropism (*Franz et al., 2014*). Deletion of *sid-3* reduces Orsay virus infection by >5 logs in vivo in the *C. elegans* host (*Jiang et al., 2017*). In the current study, infection of mice lacking TNK2 by oral gavage, the analogous route of infection as for Orsay virus infection of *C elegans*, led to increased survival of animals deficient in TNK2. These results clearly demonstrate the functional parallels between *sid-3* and *TNK2* in their respective hosts in vivo. Thus, our study has identified another example wherein novel insight in mammalian biology emerge from initial discoveries in model organism studies.

## Materials and methods

**Key resources table**

| Reagent type (species) or resource | Designation | Source or reference | Identifiers | Additional information |
|---|---|---|---|---|
| Gene (human) | TNK2 isoform1 | GenBank | NM_005781.4 | |
| Gene (human) | TNK2 isoform2 | GenBank | BC028164.1 | |
| Gene (human) | TNK2 isoform3 | GenBank | NM_001308046.1 | |
| Gene (human) | WASL | GenBank | NM_003941.3 | |
| Gene (human) | NCK1 | GenBank | NM_006153.5 | |

*Continued on next page*

*Continued*

| Reagent type (species) or resource | Designation | Source or reference | Identifiers | Additional information |
|---|---|---|---|---|
| Strain, strain background (Mouse) | C57BL/6J | The Jackson Laboratory | 000664 | |
| Strain, strain background (Mouse) | C57BL/6J TNK2 KO | This paper | | Generated at the Genome Engineering and iPSC Center (GEiC) at Washington University. David Wang lab. |
| Strain, strain background (Virus) | EMCV VR-129 strain | Michael Diamond lab | | Michael Diamond lab |
| Strain, strain background (Virus) | Coxsackie B3 Virus Nancy strain | Julie Pfeiffer lab | | Julie Pfeiffer lab |
| Strain, strain background (Virus) | Poliovirus Mahony strain | Nihal Altan-Bonnet lab | | Nihal Altan-Bonnet lab |
| Strain, strain background (Virus) | Adenovirus A5 | David Curiel lab | | David Curiel lab |
| Strain, strain background (Virus) | Influenza A virus WSN strain | Adrianus Boon lab | | Adrianus Boon lab |
| Strain, strain background (Virus) | enterovirus D68 | ATCC | ATCC VR-1826 | |
| Strain, strain background (Virus) | Parainfluenza | Robert A Lamb lab | | Robert A. Lamb lab |
| Strain, strain background (Virus) | GFP-EMCV | Frank JM van Kuppeveld lab | | Frank JM van Kuppeveld lab |
| Strain, strain background (Virus) | GFP-CVB3 | Frank JM van Kuppeveld lab | | Frank JM van Kuppeveld lab |
| Cell line (human) | A549 | ATCC | ATCC CCL-185 | |
| Cellline (human) | A549 *TNK2* KO1 | This paper | | Generated by CRISPR at David Wang lab. |
| Cell line (human) | A549 *TNK2* KO2 | This paper | | Generated by CRISPR at David Wang lab. |
| Cell line (human) | A549 *WASL* KO | This paper | | Generated by CRISPR at David Wang lab. |
| Cellline (human) | A549 *TNK2 WASL* dKO | This paper | | Generated by CRISPR at David Wang lab. |
| Cell line (human) | A549 *TNK2 NCK1* dKO | This paper | | Generated by CRISPR at David Wang lab. |

*Continued*

| Reagent type (species) or resource | Designation | Source or reference | Identifiers | Additional information |
|---|---|---|---|---|
| Cell line (human) | A549 *WASL NCK1* dKO | This paper | | Generated by CRISPR at David Wang lab. |
| Cell line (human) | A549 *TNK2 WASL NCK1* tKO | This paper | | Generated by CRISPR at David Wang lab. |
| Cell line (human) | Hela | ATCC | ATCC CCL-2 | |
| Cellline (human) | 293T | ATCC | ATCC CRL-3216 | |
| Cell line (human) | Hap1 WT | Horizon | C631 | |
| Cellline (human) | Hap1 *TNK2* KO | Horizon | HZGHC002454c026 | |
| Cell line (human) | Hap1 *WASL* KO | Horizon | HZGHC002632c003 | |
| Cellline (human) | RD (rhabdomyosarcoma) | ATCC | ATCC CCL-136 | |
| Cell line (mouse) | C57BL/6J primary lung fibroblast | This paper | | Generated by lung digestion at David Wang lab. |
| Cell line (mouse) | C57BL/6J *Tnk2* KO primary lung fibroblast | This paper | | Generated by lung digestion at David Wang lab. |
| Cellline (hamster) | BHK-21 | ATCC | ATCC CCL-10 | |
| Antibody | Mouse monoclonal anti-TNK2 clone A11 | Santa Cruz | sc-28336 | WB (1:1000) |
| Antibody | Rabbit monoclonal anti-WASL | Abcam | ab126626 | WB (1:1000) Used in *Figure 2A*, *Figure 3—figure supplement 2A,C,D* |
| Antibody | Rabbit polyclonal anti-WASL | Sigma | HPA005750 | WB (1:1000) Used in *Figure 6C* |
| Antibody | Rabbit polyclonal anti-NCK1 | Millipore | 06–288 | WB (1:1000) |
| Antibody | Mouse monoclonal anti-actin | Sigma | MAB1501 | WB (1:1000) |
| Antibody | Mouse polyclonal anti-EMCV antibodies | This paper | | Antibodies generated in Michael Diamond lab. ICC (1:1000) |
| Antibody | Mouse monoclonal anti-Coxsakie B3 virus | ThermoFisher | MAB948 | ICC (1:1000) |
| Antibody | Mouse monoclonalanti-Avenovirus A5 | ThermoFisher | MA5-13643 | ICC (1:1000) |
| Antibody | Mouse Monoclonal anti-influenza virus NP | Millipore | MAB8258B | ICC (1:2000) |
| Antibody | Rabbit polyclonal anti-Enterovirus D68 | GeneTex | GTX132313 | ICC (1:1000) |
| Antibody | Mouse monoclonal anti-poliovirus antibodies | Nihal Altan-Bonnet lab | | Antibodies generated in Nihal Altan-Bonnet lab. ICC (1:2000) |
| Antibody | Mouse monoclonal anti-HA | ThermoFisher | 26183 | WB (1:1000) |

*Continued on next page*

*Continued*

| Reagent type (species) or resource | Designation | Source or reference | Identifiers | Additional information |
|---|---|---|---|---|
| Antibody | Mouse monoclonal anti-Flag | GenScript | A00187-100 | WB (1:1000) |
| Antibody | Mouse monoclonal anti-c-Myc | Invitrogen | 13–2500 | WB (1:1000) |
| Antibody | Mouse monoclonal anti-double stranded J2 antibodies | Scicons | 10010200 | IFA (1:500) |
| Recombinant DNA reagent | pReceiver-TNK2 | Genecopia | EX-Y4392-M02 | |
| Recombinant DNA reagent | pReceiver-WASL | Genecopia | EX-I2067-M68 | |
| Recombinant DNA reagent | Lenti CRISPR v2 | Addgene | 98290 | |
| Recombinant DNA reagent | pSPAX2 | Addgene | 12260 | |
| Recombinant DNA reagent | pMD2.G | Addgene | 12259 | |
| Recombinant DNA reagent | pCW57-GFP-P2A-MCS (Neo) | Addgene | 89181 | |
| Recombinant DNA reagent | pFCIV | PMID: 27384652 | | Michael Diamond lab |
| Recombinant DNA reagent | C5V | Addgene | 26394 | |
| Recombinant DNA reagent | pcDNA-NCK1 | Genscript | OHu24619D | |
| Recombinant DNA reagent | pcDNA-TNK2 isoform 3 | Genescript | OHu13497C | |
| Sequence-based reagent (primers and Oligonucleotides) | All cloning primers and oligonucleotides used are in supplemental table | This paper | | Synthesized by IDT. David Wang lab. |
| Peptide, recombinant protein | Alt-R S.p. Cas9 nuclease 3NLS protein | IDT | 1081058 | |
| Commercial assay or kit | BCA assay | ThermoFisher | 23235 | |
| Commercial assay or kit | Cell titer-glo assay | Promega | G7570 | |
| Commercial assay or kit | QuikChange II Site-Directed Mutagenesis Kits | Agilent | 200521 | |
| Commercial assay or kit | TaqMan Fast Virus 1-Step Master Mix | ThermoFisher | 4444434 | |
| Commercial assay or kit | Lipofectamine CRISPR MAX | ThermoFisher | CMAX00001 | |
| Commercial assay or kit | CoIP | Pierce | 88804 | |
| Commercial assay or kit | Zymo RNAeasy Miniprep | Zymo research | R2052 | |
| Commercial assay or kit | Zymo RNAeasy 96-well extraction | Zymo research | R2056 | |

*Continued on next page*

*Continued*

| Reagent type (species) or resource | Designation | Source or reference | Identifiers | Additional information |
|---|---|---|---|---|
| Commercial assay or kit | Nap-5 desalting column | GE Healthcare | GE17-0853-01 | |
| Chemical compound, drug | Aim-100 | Apexbio | 4946 | |
| Chemical compound, drug | Wiskotstatin | Sigma | 681525 | |
| Chemical compound, drug | a-sarcin | Santa Cruz | CAS 86243-64-3 | |
| Chemical compound, drug | METHIONINE,L-[35S]- | Perkin Elmer | NEG009T005MC | |
| Chemical compound, drug | CYSTEINE, L-[35S]- | Perkin Elmer | NEG022T005MC | |
| Chemical compound, drug | Alexa FluorA647 succinimidyl ester | ThermoFisher | A20106 | |
| Chemical compound, drug | Polybrene | Millipore | TR-1003-G | |
| Chemical compound, drug | Puromycin | Sigma | P8833 | |
| Chemical compound, drug | Doxycyclin | Sigma | D1822 | |
| Chemical compound, drug | Dynasore | Sigma | D7693 | |
| Chemical compound, drug | Pitstop-2 | Sigma | SML1169 | |
| Chemical compound, drug | CK-869 | Sigma | C9124 | |
| Chemical compound, drug | Pirl1 | Hit2leads | SC-5137877 | |
| Software, algorithm | Flowjo V10 | FlowJo, LLC | | Commercial software for flow cytometry analysis |
| Software, algorithm | Prism V7 | GraphPad Software, Inc | | Commercial software for statistical analysis |
| Software, algorithm | Volocity V6.3 | PerkinElmer | | Commercial software for image analysis |

## Cell culture and viruses

A549 cells were cultured and maintained in DMEM supplemented with 25 mM HEPES, 2 mM L-glutamine, 1X non-essential amino acids, 10% Fetal bovine serum (FBS) and 100 u/ml antibiotics (penicillin and streptomycin). 293T (human embryonic kidney), BHK-21 (Baby hamster kidney), RD (rhabdomyosarcoma) and Hela cells were cultured and maintained in DMEM with 10% FBS. Haploid cells (HAP1) were cultured and maintained in IMDM with 10% FBS. Mouse primary lung fibroblasts were isolated by lung digestion as described (*Edelman and Redente, 2018*). Primary lung fibroblasts were derived from a *Tnk2* homozygous knockout mouse (generated at the Genome Engineering and iPSC Center (GEiC) at Washington University as described below) and a homozygous wild type littermate and were cultured in DMEM with 20% FBS. All cell lines used in this study were tested negative for mycoplasma and their identity was indicated in the key resource table. Viruses

were obtained from the following: EMCV VR-129 strain (Michael Diamond), coxsackie virus B3 Nancy strain (Julie Pfeiffer), poliovirus Mahoney strain (Nihal Altan-Bonnet), influenza A virus WSN strain (H1N1) (Adrianus Boon), adenovirus A5 (David Curiel), enterovirus D68 (ATCC), parainfluenza virus (Robert A. Lamb), GFP-EMCV and GFP-CVB3 (Frank J. M. van Kuppeveld). EMCV was amplified on BHK-21 cells; CVB3 and polio viruses were amplified on HeLa cells; and enterovirus D68 was amplified on RD cells.

## Reagents and antibodies

α-sarcin was purchased from Santa Cruz. $^{35}$S methionine and $^{35}$S cysteine were purchased from Perkin Elmer. Alexa FluorA647 succinimidyl ester was purchased from ThermoFisher Scientific. Inhibitors were purchased from commercial vendors as follows: Aim-100 (Apexbio), Wiskostatin (Sigma), Dynasore (Sigma), Pitstop-2 (Sigma), CK-869 (Sigma), Pirl1 (Hit2leads). Anti-EMCV mouse polyclonal antibodies were provided by Michael Diamond. Anti-poliovirus antibodies were provided by Nihal Altan-Bonnet. Other antibodies were obtained from commercial vendors as follows: Anti-coxsackie virus B3 antibodies (ThermoFisher), Anti-adenovirus A5 antibodies (ThermoFisher), Anti-influenza A virus NP antibodies (Millipore), Anti-TNK2 (A11) (Santa Cruz), Anti-WASL (Abcam and Sigma), Anti-actin, clone C4 (Sigma), Anti-NCK1 (Millipore), enterovirus D68 Ab (GeneTex), Anti-HA (ThermoFisher), Anti-Flag (GenScript), Anti-c-Myc (Invitrogen), Anti-double stranded J2 antibodies (Scicons).

## Plasmid constructions

All primers and oligonucleotides used in this study are listed in the supplementary table. Single guide RNA (sgRNA) oligonucleotides were synthesized by Integrated DNA Technologies (IDT). sgRNA oligoes were annealed and cloned into Lenti-CRISPR V2 plasmid digested by BsmBI. TNK2 (pReceiver-TNK2) and WASL (pReceiver-WASL) ORF clones were obtained from GeneCopoeia. NCK1 (pcDNA-NCK1) ORF clone was obtained from GenScript. Mutations in the sgRNA binding sites were introduced by site-directed mutagenesis (Stratagene) according to the manufacturer's protocol. TNK2, WASL and NCK1 were subcloned into a Lentivirus expression vector pFCIV digested with Asc1 and Age1. The FRET control plasmid C5V was obtained from Addgene. mVenus cassette was subcloned into pcDNA-NCK1 and pReceiver-WASL. mCerulean cassette was subcloned into pReceiver-TNK2. Myc tagged TNK2 and HA tagged WASL was generated by annealing of oligonucleotides and then subcloned into the expression vector. GFP tagged TNK2 and WASL were subcloned into a tetracycline promoter driven Lentivirus vector pCW57. Constitutive active WASL constructs were generated by site-directed mutagenesis. Domain truncation was generated by overlapping PCR. In brief, to generate pFCIV-WASL ΔWH1, pFCIV-WASL ΔB, pFCIV-WASL ΔPRD, pFCIV-WASL ΔGBD, and pFCIV-WASL ΔA, WASL fragments were amplified from the start of the gene to the start of the truncation and from the end of the truncation to the end of the gene, and the fragments were then joined by overlapping PCR. The resulting product was digested by AgeI and AscI and ligated into Lentivirus expression vector pFCIV that were cut by the same restriction enzymes.

## Lentivirus production and cell transduction

800 ng of Lenti CRISPR V2 plasmids or pFCIV plasmids or pCW57 plasmids were transfected with 800 ng pSPAX2 and 400 ng pMD2.G into 293 T cells using Lipofectamine 2000 according to the manufacturer's protocol. Cell culture supernatant was harvested two days post transfection and stored at −80°C. For transduction, A549 cells were seeded one day before transduction. Cells were spin transfected with lentivirus and 8 µg/ml of polybrene at MOI 5. Two days after transduction, cells were passaged and either selected under corresponding antibiotics or fluorescently sorted through flow cytometry.

## CRISPR genome editing

A549 naïve cells were transduced by lentiviruses that express the corresponding sgRNA and Cas9 protein. Transduced cells were passaged two days later and then selected with 2 µg/ml puromycin for 7 days. Cells were passaged once during this selection. Clonal selection was performed through limiting dilution in 96-well plates. After two weeks, single cell clones were picked and expanded in 24-well plates. The desired genome editing was identified by a restriction enzyme digestion-based genome typing assay. Genome edited clonal cells were further sequenced by Sanger sequencing to

define the precise genome editing event. Detection of a homozygous 28 bp (base pair) deletion allele of *TNK2* defined *TNK2* KO1 and detection of a 25 bp deletion allele and 137 bp deletion allele of *TNK2* defined *TNK2* KO2. Identification of a 1 bp insertion allele and a 2 bp deletion allele of *WASL* defined *WASL* KO. An insertion allele of 94 bp in *NCK1* was observed and this defined *NCK1* KO. *TNK2* KO1, *WASL* KO and *NCK1* KO were used in all experiments except places specified using other cells.

For homologous template-directed DNA repair through CRISPR genome editing, assembled CRISPR RNP were transfected into A549 cells with single stranded oligodeoxynucleotide (ssODN) as described (*Jacobi et al., 2017*). In brief, Alt-R S.p. Cas9 nuclease 3NLS protein, Alt-R CRISPR-Cas9 crRNA designed to target the deleted *TNK2* KO1 genomic region, Alt-R CRISPR-Cas9 ATTO 550 tagged tracrRNA and ssODN were purchased from IDT. A549 cells were seeded into 12-well plates one day before transfection. Equal molar crRNA and tracrRNA were annealed by heating to 95℃ for 5 min and then cooled to room temperature. RNP was assembled by combing equal molar ratio of annealed cr-tracrRNA with Cas9 nuclease protein in opti-MEM. RNP was then transfected with ssODN into A549 cells by lipofectamine CRISPR-MAX according to the manufacturer's protocol. 24 hr after transfection, ATTO 550 positive cells were FACS sorted individually into 96-well plates. Single cell colonies were expanded one week after sorting and genotyped with a restriction enzyme-based genotyping assay. Template-directed DNA repair was finally confirmed by both Sanger and PCR product deep sequencing.

For generation of gene double or triple knockout cells, the same CRISPR RNP transfection method was used as homologous template-directed DNA repair using corresponding crRNA designed to target genes with ssODN omitted. A double knockout of *NCK1* and *WASL* was generated by CRISPR-Cas9 genome editing of the *NCK1* locus in the *WASL* KO cells. A double knockout of *WASL* and *TNK2*, a double knockout of *NCK1* and *TNK2*, and a triple knockout of *WASL*, *NCK1* and *TNK2* were generated by CRISPR-Cas9 genome editing of the *WASL* locus and *NCK1* locus in the *TNK2* KO1 cells. Genome editing events were screened by a restriction enzyme digestion-based genotyping assay.

## EMCV virus labeling and infection for imaging

EMCV was amplified in BHK-21 cells and purified according to previous publications (*Brandenburg et al., 2007*; *Staring et al., 2017*). Virus labeling was performed with Alexa FluorA647 succinimidyl ester in a 1:10 molar ratio and was then purified through Nap-5 desalting column (GE Healthcare). Labeled viruses were aliquoted and stored at −80℃. For EMCV entry imaging analysis, A549 cells were seeded at 3000 cells per well in an 18-well IBD imaging slide chamber one day before infection. The next day, cells were washed once with serum free DMEM and then inoculated with labeled EMCV virus at an MOI of 20 on ice for one hour. Cells were washed three times with ice-cold PBS after on ice binding. Cells were then fixed with 4% paraformaldehyde for 15 min at room temperature or cells were switched to 37℃ incubation with complete medium for internalization. After incubation with complete medium for 30 min, cells were then washed once with PBS and fixed with 4% paraformaldehyde. Fixed cells were mounted in IBD mounting medium for image analysis. For GFP-TNK2 and GFP-WASL localization with labeled EMCV virus imaging, live cell experiment was performed in an 18-well IBD imaging slide in a temperature-controlled imaging chamber.

## EMCV virus binding and internalization assay

A549 cells were seeded at $1 \times 10^5$ cell per well in a 24-well plate one day before the assay. Cells were chilled on ice for 30 min and then washed with ice-cold DMEM before inoculation. EMCV was diluted in ice-cold DMEM with 0.1% BSA and then inoculated at an MOI of 20 in 250 µl of DMEM per well of 24-well plate. Viruses were allowed to bind on ice for one hour. For virus binding experiments, cells were washed three times with ice-cold PBS and then lysed in 350 µl Trizol reagent. For virus internalization assay using trypsinization, experiments were performed as described previously for WNV and AAV (*Berry and Tse, 2017*; *Hackett et al., 2015*). In brief, after one hour of binding, the virus inoculum was removed and pre-warmed complete medium was added onto cells. Cells were incubated in a 37℃ water bath for 30 min to allow for virus internalization. After incubation, cells were washed three times with ice-cold PBS and then trypsinized for 6 min to remove surface

bound virus. Trypsinized cells were then washed again three times and then spin at 300 g for 5 min. Cell pellets were lysed in 350 µl Trizol reagents. RNA was extracted using the 96-well Zymo RNA easy column extraction according to the manufacturer's protocol. Viral RNA was quantified by one step reverse transcription quantitative real-time PCR with an EMCV assay probe (Primers: forward 5'-CGATCACTATGCTTGCCGTT-3'; reverse 5'-CCCTACCTCACGGAATGGG-3'; Taqman probe 5' FAM-AGAGCCGATCATATTCCTGCTTGCCA-3'). Fold change was converted from delta delta Ct of an internal control assay by RPLP0 (Ribosomal protein lateral stalk subunit P0). For labeled EMCV, internalization was performed the same way, after trypsinization and washing, cells were fixed in 4% paraformaldehyde for 15 min at room temperature and then washed twice with P2F (PBS with 2% Fetal bovine serum). Cells were finally resuspended in P2F and analyzed by MACS flow cytometry.

## FACS assay

A549 cells were seeded one day before infection into 96-well plates. Approximately, 16 hr after seeding, cells were infected by EMCV at an MOI of 1. One hour after infection, the inoculum was removed and cells were cultured in DMEM with 2% FBS. 10 hr post infection, cells were trypsinized and fixed with 4% paraformaldehyde. Fixed cells were then permeabilized with perm buffer (1 g Saponin, 10 ml HEPES, 0.025% Sodium Azide in 1L HBSS) for 15 min. After permeabilization, cells were incubated with primary antibodies for one hour and then washed twice before incubation with fluorescently conjugated secondary antibodies. After one hour of secondary antibody incubation, cells were washed three times with perm buffer and then resuspended with 70 µl of FACS buffer P2F (PBS with 2% fetal bovine serum). Infected cells were then analyzed and quantified through MACS flow cytometry (Miltenyi Biotec). FACS analysis of infection by CVB3, polio, enterovirus D68, adenovirus and influenza virus on either A549 or Hap1 cells were performed the same as EMCV infection, except that cells were harvested 8 hr post infection for CVB3, adenovirus, enterovirus D68 and influenza virus, and 6 hr post infection for poliovirus.

## Multistep growth analysis for EMCV and Polio virus

For multistep growth analysis, A549 or Hap1 cells were infected by EMCV at an MOI of 0.01. One hour after inoculation, cells were washed five times with serum free DMEM and were then cultured in DMEM or IMDM with 2% FBS. Culture supernatant was collected at time 0, 6, 12, 24, 36, and 48 hr post infection. Viruses released in the culture supernatant were titrated on BHK-21 cells by plaque assay. For polio virus multi-step growth titration, A549 or Hap1 cells were infected by virus at MOI 0.01 and culture supernatant were collected. Released viruses were titrated on HeLa cells by plaque assay. For Coxsackie B3 virus multi-step growth titration, A549 cells were infected by virus at MOI 0.01 and culture supernatant were collected at time 0, 6, 12, 24, 36, and 48 hr post infection. Released viruses were titrated on HeLa cells by plaque assay.

## EMCV genomic RNA transfection

EMCV genomic RNA was extracted from pelleted virions by phenol chloroform extraction. For RNA transfection, A549 cells (gene knockouts and control) were seeded into 24-well plate at $1 \times 10^5$ cells per well. 16 hr after seeding, cells were transfected with 1.6 µg EMCV RNA by Lipofectamine 3000 according to the manufacturer's protocol. Cell culture supernatant was collected at 10 hr post transfection and was then titrated on BHK-21 cells by plaque assay.

## Co-immunoprecipitation and western blot

Flag C-terminal tagged NCK1 was transfected with HA N-terminal tagged WASL or with Myc N-terminal tagged TNK2 into 293 T cells. 48 hr after transfection, cells were lysed in IpLysis buffer (Invitrogen). Protein concentrations were quantified by BCA assay. 500 µg protein lysates were incubated with 2 µg of anti-flag antibodies at 4°C for overnight for immunoprecipitation. After antibody binding, the immuno-complex was incubated with protein A/G magnetic beads for one hour at room temperature. The beads were washed and proteins bound to antibodies were eluted according to the manufacture's protocol (Invitrogen). Protein samples were prepared with 4X NuPAGE sample buffer (Invitrogen) and then resolved on 4–12% NuPAGE gel. Proteins were transferred to a PVDF membrane and then blocked with 5% skim milk. Primary antibodies were incubated in 5% milk for overnight at 4°C. After three washes with PBST (PBS with 0.3% of tween-20), corresponding HRP

conjugated secondary antibodies were incubated in 5% milk at room temperature for 1 hr. The membrane was washed five times with PBST and was then developed with chemiluminescent substrate for 5 min and then imaged using a Bio-Rad chemiluminescence imager.

## α-Sarcin pore forming assay

A549 cells were seeded at $3.6 \times 10^4$ per well of a 48-well plate. The next day, cells were washed once with PBS and then incubated with methionine and cysteine free DMEM for one hour. Subsequently, cells were inoculated with EMCV virus at an MOI of 50 on ice for one hour. Inoculum were then removed and cells were washed three times with ice cold PBS. Cells were then incubated with 100 µg/ml α-sarcin diluted in methionine and cysteine free DMEM for 90 min at 37°C. After α-sarcin treatment, cells were pulsed with 7.4 MBq $^{35}$S methionine and $^{35}$S cysteine in DMEM for 20 min. Next, cells were washed four times with ice cold PBS and then lysed with SDS sample buffer. Proteins were resolved on 7.5% SDS-PAGE gel and then dried on a Bio-Rad gel dryer. Dried gels were exposed to a phosphorimager overnight and then scanned by Fuji IFA imaging system. As an internal loading control, a parallel SDS-PAGE gel was run and then proteins were transferred to PVDF membrane and blotted with anti-actin antibodies.

## Immunofluorescence assay

For EMCV co-localization with EEA1 marker, A549 cells were seeded at 3000 cells per well in an 18-well IBD imaging slide chamber one day before infection. The next day, cells were washed once with serum free DMEM and then inoculated with labeled EMCV virus at an MOI of 20 on ice for one hour. Cells were washed three times with PBS after on ice binding. Cells were then switched to 37°C incubation with complete medium for internalization at indicated time. After incubation with complete medium for different time, cells were then washed once with PBS and fixed with 4% paraformaldehyde for 10 min. Fixed cells were then permeabilized with 0.2% saponin for 10 min at room temperature and subsequently blocked with IFA blocking buffer (10% goat serum, 0.05% saponin in PBS) at room temperature for 30 min. Cells were then incubated with EEA1 antibodies at 1:100 dilution in blocking buffer overnight with shaking. Cells were washed three times with PBS and then incubated with secondary antibodies at room temperature for 1 hr. After incubation, cells were washed once with PBS and then incubated with Hoechst at 1:1000 dilution for 10 min at room temperature. Cells were washed three time with PBS and mounted in IBD mounting medium for image analysis. For dsRNA immunostaining in EMCV infected cells, the same procedure was performed as EEA1 immunostaining except that 0.1% Triton was used instead of Saponin.

## Confocal imaging and FRET analysis

Cells on slides or in an IBD imaging slide chamber were examined on a Zeiss airy scan confocal microscope (LSM 880 II). A Plan Apochromat 63X, 1.4-numerical-aperture oil objective lens (Carl Zeiss, Germany) was used to image labeled virus infection. For FRET analysis, 293 T cells were seeded on coverslips and transfected with FRET pairs, mCerluean tagged TNK2 with mVenus tagged NCK1 and mCerulean tagged WASL with mVenus tagged NCK1 respectively. 24 hr post transfection, cells were fixed in 4% paraformaldehyde and mounted on slides. Cells were imaged on a Zeiss airy scan confocal with a Plan Apochromat 100X, 1.4-numerical-aperture oil objective lens. Acceptor photo bleach was performed with 80% laser intensity of the imaging channel. Images were taken before and after photo bleach and FRET efficiency were calculated after image acquisition on Zen pro software (Carl Zeiss, Germany). For quantification of colocalization, image analysis was performed using Volocity software V6.3 (PerkinElmer). In brief, virus pixels were automatically detected in the red channel by Volocity with a particle size filter of no less than 25 pixels, colocalization were detected by intersection pixels of the red channel and green channel. Percent of colocalization was determined by dividing the colocalized intersection pixels by the total virus pixels. Pearson correlation coefficient of EMCV and EEA1 was determined for each image using Volocity by setting a threshold of each channel above background.

## Transmission electron microscopy

EMCV was incubated with A549 cells (wild type, *TNK2* KO and *WASL* KO) at an MOI of 20 for 1 hr on ice. Cells were then washed with serum free DMEM and incubated with A549 culture media (2%

FBS) for 6 hr. Infected cells were washed with PBS and fixed with 2% paraformaldehyde, 2.5% glutaraldehyde (Polysciences Inc, Warrington, PA) in 100 mM cacodylate buffer for 1 hr at room temperature. Next, cells were scraped from plates using a rubber cell scraper and cell pellets were embedded in agarose. Agarose embedded cell pellets were post-fixed in 1% osmium tetroxide (Polysciences Inc) for 1 hr, then rinsed extensively in dH20 prior to *en bloc* staining with 1% aqueous uranyl acetate (Ted Pella Inc, Redding, CA) for 1 hr. Following several rinses in dH20, samples were dehydrated in a graded series of ethanol and embedded in Eponate 12 resin (Ted Pella Inc). Sections of 95 nm were cut with a Leica Ultracut UCT ultramicrotome (Leica Microsystems Inc, Bannockburn, IL), stained with uranyl acetate and lead citrate, and viewed on a JEOL 1200EX transmission electron microscope (JEOL USA, Peabody, MA) equipped with an AMT eight mega-pixel digital camera (Advanced Microscopy Techniques, Woburn, MA).

## In vivo infection experiments

Animal experiments were conducted under the supervision of Department of Comparative Medicine at Washington University in St. Louis. All animal protocols were approved by the Washington University Institutional Animal Care and Use Committee (Protocol #20170194 and #20180289). *Tnk2* knockout mice were generated in the C57BL/6 background by CRISPR-Cas9 genome editing at the Genome Engineering and iPSC Center (GEiC) at Washington University. All animals were housed in the pathogen-free barrier. Age-matched animals with mixed gender (6–8 weeks old, 5 females and six males for *Tnk2* knockout, 2 females and five males for wild type littermates) were infected with $1 \times 10^7$ PFU of EMCV (two doses at day 0 and day 1) via oral gavage according to previous publication (*Wang et al., 2015*). Infected mice were monitored for 3 weeks for all experiments. For virus titration in infected mouse tissues, whole brain and heart were harvested at day four post virus gavage (16 *Tnk2* knock out and 14 wild type male mice). Tissues were weighed and a 1:5 wt to volume ratio of PBS was added to tissue sample. Tissues were homogenized by bead beating for 1 min on a MagNALyzer bead beater, cool down for 2 min on ice and then another minute for bead beating. Tissue homogenates were centrifuged at 13,000 g for 5 min and supernatant were collected for plaque assay on BHK-21 cells.

## Statistical analysis

For statistical analysis, Student T-test was performed on the average values from three replicates. All data shown with statistics are representative of at least two independent experiments. Log-rank test and Mann Whitney test were done in Graphpad Prism V7. Statistical significance is indicated as below: NS: no statistical significance, *: $p<0.05$, **: $p<0.01$, ***: $p<0.001$, ****: $p<0.0001$, *****: $p<0.00001$.

## Acknowledgements

This work was supported in part by National Institutes of Health R01 AI134967.

We thank Adrianus Boon, Skip Virgin, Celeste Morley and John Cooper for helpful discussions. We thank Rong Zhang and Michael S Diamond for access to FACS instrumentation. FRET experiments were performed at the Washington University Center for Cellular Imaging (WUCCI) supported by Washington University School of Medicine, The Children's Discovery Institute of Washington University and St. Louis Children's Hospital (CDI-CORE-2015–505) and the Foundation for Barnes-Jewish Hospital (3770). We thank Wandy Beatty for assistance with electron microscopy and laser scanning confocal. We thank the Genome Engineering and iPSC Center (GEiC) at the Washington University in St. Louis for their sgRNA validation and genotyping services for generating the *Tnk2* knockout mouse. We thank the flow cytometry core at Department of Pathology and Immunology, Washington University School of Medicine for assisting cell sorting. We thank Tim Schaff and Darren Kreamalmeyer for assistance with mouse breeding.

# Additional information

### Funding

| Funder | Grant reference number | Author |
|---|---|---|
| National Institutes of Health | R01 AI134967 | David Wang |

The funders had no role in study design, data collection and interpretation, or the decision to submit the work for publication.

### Author contributions

Hongbing Jiang, Conceptualization, Resources, Data curation, Software, Formal analysis, Supervision, Validation, Investigation, Visualization, Methodology, Project administration; Christian Leung, Data curation, Formal analysis, Validation, Investigation, Methodology; Stephen Tahan, Data curation, Software, Formal analysis, Validation, Investigation, Methodology; David Wang, Conceptualization, Resources, Data curation, Formal analysis, Supervision, Funding acquisition, Project administration

### Author ORCIDs

Hongbing Jiang (iD) https://orcid.org/0000-0002-7320-3349
David Wang (iD) https://orcid.org/0000-0002-0827-196X

### Ethics

Animal experimentation: Animal experiments were conducted under the supervision of Department of Comparative Medicine at Washington University in St. Louis. All animal protocols were approved by the Washington University Institutional Animal Care and Use Committee (Protocol #20170194 and #20180289).

### Decision letter and Author response

Decision letter https://doi.org/10.7554/eLife.50276.sa1
Author response https://doi.org/10.7554/eLife.50276.sa2

# Additional files

### Supplementary files

• Supplementary file 1. Primers and oligonucelotides used in this study.

• Transparent reporting form

### Data availability

All data generated or analysed during this study are included in the manuscript and supporting files. Source data files were provided.

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
