## [Decision Letter]

Thank you for submitting your article "Entry by multiple picornaviruses is dependent on a pathway that includes TNK2, WASL and NCK1" for consideration by *eLife*. Your article has been reviewed by three peer reviewers, and the evaluation has been overseen by a Reviewing Editor and Karla Kirkegaard as the Senior Editor.

The reviewers have discussed the reviews with one another and the Reviewing Editor have invited you to prepare a revised submission.

The work describes the detailed characterization of the role of three mammalian homologues (TNK2, WASL and NCK1) of genes previously found by the authors in a genetic screen for viral susceptibility factors in *C. elegans*. Knockout of these genes leads to reduced levels of infection by diverse picornaviruses. The authors pinpoint the defect in the viral entry phase after binding and pore formation. Finally, in a mouse of EMCV infection, reduced mortality was found in TNK2 knockout mice. The manuscript is well-written, the experimental approach is technically sound and the data generally support the conclusions. Understanding cellular components that support infection by the important picornavirus family is of high interest to a broad field of biologists. We would like the authors to address several major areas that could be improved:

Essential revisions:

1) Animal studies: The effects of knockout of the three genes ranges from a moderate effect to a very small effect (albeit statistically significant). This effect can vary depending on the percentage infected cells at different MOI's. For example at higher infection ratio's the effect of WASL-KO for CV-B3 (Figure 1—figure supplement 1B) is negligible, while there might be an effect at lower MOIs. By plotting the data as relative infection compared to WT control (as is done throughout the manuscript), important information is lost regarding the absolute infection levels of (WT and KO) cells. It would be beneficial to plot the data as absolute infection percentages.

In addition, the authors state that expression of TNK2 induced IFNs, which would suggest that this factor plays an important role in antiviral signaling. Therefore, changes in survival may be related to alterations in some aspect of immune activation/signaling. This should be addressed.

Further, in the mouse model the experiment was stopped at day 10, while there was still a mouse dying at day 9 in the KO group. There is a concern that KO results in a delay in mortality but, if taken out longer, not in significantly reduced overall mortality.

For Figure 7 to be more impactful, the authors should provide titer information from various tissues isolated from infected animals (two reviewers suggested this).

2) The impact of deletion of TNK2 and WASL is quite modest. It would therefore seem to be an overstatement to suggest that these factors regulate the internalization of all picornaviruses when the most striking data seems to be with EMCV. Growth curves for infectious viral titers for the other viruses included in the manuscript would bolster the authors' conclusions. Moreover, I think it is an overstatement to call the genes "critical" for infection, especially in the Abstract. There is no doubt that they are "important" and of interest but some of the more modest effects do not justify the word "critical".

3) The internalization assay shown in Figure 4D is lacking controls. As this assay relies on the removal of bound particles by trypsin, it would be important to show that this type of assay can be applied to picornaviruses, which are often quite resistant to this treatment.

4) It is surprising that the reduction of infection is so modest in fibroblasts isolated from KO animals (Figure 7C). These data could be bolstered by inclusion of titer data.

---

## [Author Response]

Essential revisions:1) Animal studies: The effects of knockout of the three genes ranges from a moderate effect to a very small effect (albeit statistically significant). This effect can vary depending on the percentage infected cells at different MOI's. For example at higher infection ratio's the effect of WASL-KO for CV-B3 (Figure 1—figure supplement 1B) is negligible, while there might be an effect at lower MOIs. By plotting the data as relative infection compared to WT control (as is done throughout the manuscript), important information is lost regarding the absolute infection levels of (WT and KO) cells. It would be beneficial to plot the data as absolute infection percentages.

As requested we have now included graphs of the absolute infection percentage as supplementary figures. We have maintained the relative infection graphs in the primary figure due to relative ease of interpretation and comparison. However, we can make those supplementary absolute graphs primary, if requested. In addition, all absolute infection data are included in the *eLife* source data file.

In addition, the authors state that expression of TNK2 induced IFNs, which would suggest that this factor plays an important role in antiviral signaling. Therefore, changes in survival may be related to alterations in some aspect of immune activation/signaling. This should be addressed.

Little is known about the link between TNK2 and interferon induction. To date, there is only one published study suggesting that TNK2 overexpression can induce interferon stimulated genes and can restrict HCV replication (Fujimoto et al., 2011), which we cited to provide relevant background information. As our EMCV phenotypes are reduced lethality and reduced viral titers in TNK2 KO, at the simplest level, these observations are not consistent with an antiviral signaling mechanism. While we are quite interested in investigating a potential antiviral function of TNK2, we believe that rigorous study of the impact on immune activation and signaling, such as crossing of TNK2 KO animals into IFN deficient mice, are beyond the scope of this manuscript.

Further, in the mouse model the experiment was stopped at day 10, while there was still a mouse dying at day 9 in the KO group. There is a concern that KO results in a delay in mortality but, if taken out longer, not in significantly reduced overall mortality.

In the mouse experiment, we monitored the infected mice for 21 days and no additional deaths occurred after day 9. We have extended the survival curve in Figure 7E to day 21 to reflect this.

For Figure 7 to be more impactful, the authors should provide titer information from various tissues isolated from infected animals (two reviewers suggested this).

The EMCV titers of infected mouse heart and brain, the tissues most commonly analyzed in EMCV infection, have been added as Figure 7F and 7G.

2) The impact of deletion of TNK2 and WASL is quite modest. It would therefore seem to be an overstatement to suggest that these factors regulate the internalization of all picornaviruses when the most striking data seems to be with EMCV. Growth curves for infectious viral titers for the other viruses included in the manuscript would bolster the authors' conclusions. Moreover, I think it is an overstatement to call the genes "critical" for infection, especially in the Abstract. There is no doubt that they are "important" and of interest but some of the more modest effects do not justify the word "critical".

As requested, we have added multi-step growth curves for CVB3 in TNK2, WASL and NCK1 knockout A549 cells (Figures 1E, 2E, 2K). We have replaced all instances of “critical” with “important” in the text.

3) The internalization assay shown in Figure 4D is lacking controls. As this assay relies on the removal of bound particles by trypsin, it would be important to show that this type of assay can be applied to picornaviruses, which are often quite resistant to this treatment.

We have added controls (Figure 4—figure supplement 1C-E) demonstrating that Trypsin treatment consistently removes the same fraction of bound EMCV in control, TNK2 KO1 and WASL KO cells.

4) It is surprising that the reduction of infection is so modest in fibroblasts isolated from KO animals (Figure 7C). These data could be bolstered by inclusion of titer data.

We have added a multi-step growth curve and EMCV titers are consistently lower in TNK2 KO lung fibroblast than WT.